# Federated Learning with Projected Trajectory Regularization

**Tiejin Chen**[*][†]                                                              *tiejin@asu.edu*
*Arizona State University*

**Yuanpu Cao**[*]                                                                 *ymc5533@psu.edu*
*Pennsylvania State University*

**Yujia Wang**[*]                                                                 *yjw5427@psu.edu*
*Pennsylvania State University*

**Cho-Jui Hsieh**                                                               *chohsieh@cs.ucla.edu*
*UCLA*

**Jinghui Chen**                                                                 *jzc5917@psu.edu*
*Pennsylvania State University*

**Reviewed on OpenReview:** *https://openreview.net/forum?id=vfCztZvcP3*

## Abstract

Federated learning enables joint training of machine learning models from distributed clients without sharing their local data. One key challenge in federated learning is to handle non-identically distributed data across the clients, which leads to deteriorated model training performance. Prior works in this line of research mainly focus on utilizing last-step global model parameters/gradients or the linear combinations of the past model parameters/gradients, which do not fully exploit the potential of global information from the model training trajectory. In this paper, we propose a novel federated learning framework with projected trajectory regularization (FedPTR) for tackling the data heterogeneity issue, which proposes a unique way to better extract the essential global information from the model training trajectory. Specifically, FedPTR allows local clients or the server to optimize an auxiliary (synthetic) dataset that mimics the learning dynamics of the recent model update and utilizes it to project the next-step model trajectory for local training regularization. We conduct rigorous theoretical analysis for our proposed framework under nonconvex stochastic settings to verify its fast convergence under heterogeneous data distributions. Experiments on various benchmark datasets and non-i.i.d. settings validate the effectiveness of our proposed framework.

## 1 Introduction

Federated Learning (FL) (McMahan et al., 2017; Konečný et al., 2016) has recently emerged as an attractive learning paradigm where a large number of remote clients (e.g., mobile devices, institutions, organizations) cooperate to jointly learn a machine learning model without data sharing. Specifically, Federated Learning works by performing local training on each client based on their own data, while having the clients communicate their local model updates to a global server (McMahan et al., 2017; Konečný et al., 2016). Enjoying privacy protection and efficient communication, Federated Learning has been widely applied in various domains such as edge computing (Wang et al., 2019), industrial engineering (Hu et al., 2018), and healthcare (Lee et al., 2018).

---

[*]Equal contribution.
[†]Work was done while Tiejin Chen was an intern at Pennsylvania State University.

As a prevalent Federated Learning algorithm, FedAvg (McMahan et al., 2017) updates the global model by averaging multiple steps of local stochastic gradient descent (SGD) updates. Though FedAvg has demonstrated empirical success on independent and identically distributed (i.i.d.) client data, when facing the more practical non-i.i.d. data distributions across the clients, it usually leads to diverging updates and severe drift between the local and the global training objective in each client, which largely deteriorates the model training performance (Khaled et al., 2020; Karimireddy et al., 2020b; Wang et al., 2020b). For instance, imagine a group of hospitals that aims to collaboratively train a cancer diagnosis model together while keeping their own patients' data private through FedAvg. Due to different imaging machines and protocols used in hospitals, the medical images usually exhibit varying visual features (e.g., distinct intensity and contrast) (Li et al., 2021c) and thus the locally trained model may easily overfit and drift away from the global model. Similarly, in the task of anomaly detection on edge devices, the data collected on each edge device may only partially cover the entire normal space due to environmental conditions (Zhu et al., 2022). In such a case, simply training models based on FedAvg may obtain a global model that collapses to certain normal modes. As a result, many normal instances would be falsely detected as anomalies (Zhu et al., 2022).

To tackle this data heterogeneity issue, a line of research mainly focuses on utilizing or regularizing with the last-step global model parameters/gradients or the linear combinations of past model parameters/gradients. Specifically, FedProx (Li et al., 2020) makes a simple modification to FedAvg by adding a proximal regularization term in the local training objective to restrict the local updates to be closer to the last-step global model. SCAFFOLD (Karimireddy et al., 2020b) reduces client drift with a control variate that linearly accumulates past gradients and corrects for the drift. FedDC (Gao et al., 2022) follows a similar idea as SCAFFOLD with a local drift term and introduces a linear gradient correction loss, attempting to further eliminate the drift at local training.

However, such methods may not be sufficient to fully exploit global information to tackle extreme data heterogeneity. In order to better extract the essential global information from the model training trajectory, we propose a novel federated learning framework with projected trajectory regularization (FedPTR), to help guide the local model update. Specifically, during the local training, FedPTR allows local clients or the server to optimize an auxiliary dataset that mimics the learning dynamics of the recent model update and then utilizes it to project the next-step model trajectory for local training regularization. To achieve the above goal and optimize the auxiliary dataset, we adopt the Matching Training Trajectories (MTT) technique (Cazenavette et al., 2022), which was originally proposed for the task of dataset distillation. Here we want to emphasize that our use of MTT is actually different from traditional dataset distillation whose goal is to synthesize a small dataset such that models trained on the synthetic dataset can achieve similar test accuracy as if trained on the actual dataset. Firstly, most works in dataset distillation cannot fit in here as they usually require access to the entire training dataset which is not practical for federated learning. Some existing works (Zhou et al., 2020; Song et al., 2022) aim to get around this by letting local clients distill their own data and send it to the server. However, this could incur extra communication costs and raise data privacy concerns. Secondly, the goal of dataset distillation is quite ambitious and usually requires obtaining multiple model training trajectories (or re-starts) in order to obtain barely satisfactory performance. In sharp contrast, here we only require the extracted auxiliary dataset to mimic the learning dynamics of recent model updates for projecting the next-step update, which is a much easier task that can be achieved with only one global model training trajectory. We summarize our contribution as follows:

- We propose a novel federated learning framework FedPTR that handles extreme data heterogeneity across local clients. The proposed method allows each client or the server to extract essential global information from the observed global model training trajectory and utilize it to project the next-step model trajectory for local training regularization.

- We provide rigorous theoretical analysis and prove that FedPTR can achieve a convergence rate of $\mathcal{O}(1/T)$ w.r.t total communication rounds $T$ under stochastic non-convex optimization settings, which matches the best existing results.

- We conduct extensive experiments on various benchmark datasets and non-i.i.d. settings to demonstrate the effectiveness of our proposed FedPTR in training real-world machine learning models.

## 2 Related Work

**Federated Learning for Heterogeneous Data.** Federated Learning (McMahan et al., 2017) has been widely applied to a wide range of clients (e.g., mobile devices, institutions, and organizations) to collaboratively train a global model without sharing their private data. However, the heterogeneous data distribution across different devices has become one of the major bottlenecks for FL algorithms. A prominent line of work addresses this by regularizing local training objectives such as FedProx (Li et al., 2020) and Fed-Dyn (Acar et al., 2021), and client drift correction via control variates such as SCAFFOLD (Karimireddy et al., 2020b), Mime (Karimireddy et al., 2020a) and FedDC (Gao et al., 2022). Beyond FedProx-style proximal regularization toward the last-step global model, accelerated proximal-point methods explore alternative or extrapolated proximal centers to speed up convergence, such as Nesterov-type momentum extrapolation in proximal point methods (Güler, 1992) and Halpern-type anchored acceleration of the proximal operator (Kim, 2021). One recent line of work incorporates sharpness-aware minimization (SAM) into federated optimization to find flatter minima and improve generalization under data heterogeneity, such as FedSAM, MoFedSAM (Qu et al., 2022), FedSMOO (Sun et al., 2023a), FedLESAM (Fan et al., 2024), FedGMT (Li et al., 2025) and FedSCAM (Rahil et al., 2025). Another recent line of work addresses data heterogeneity by adopting server-level or client-level optimization methods such as MOON (Li et al., 2021a), FedSpeed (Sun et al., 2023b), FedPD (Zhang et al., 2020) and AFGA (Wang & Chen, 2024). Moreover, personalized federated learning (Fallah et al., 2020; Smith et al., 2017; Deng et al., 2020; Li et al., 2021b) has also been explored to better adapt global models to heterogeneous local distributions. Moreover, instead of aggregating parameters, aggregating class prototypes and representations offers natural benefits in addressing the data heterogeneity challenge, as exemplified by FedProto (Tan et al., 2022), FedNH (Dai et al., 2023), FedProtoKD (Siddika et al., 2025) and ProtoNorm (Lee et al., 2025). More recently, the shift from training small models to training and fine-tuning large generative models has reshaped which aspects of data heterogeneity matter. For example, a line of work (Yi et al., 2023; Cho et al., 2024; Bai et al., 2024; Bian et al., 2025) applies PEFT-style methods such as LoRA (Hu et al., 2022) to federated fine-tuning of LLMs, while another line (Mei et al., 2024; Morafah et al., 2024; Peng et al., 2025; Zhang et al., 2025) studies federated diffusion models under heterogeneous data.

**Dataset Distillation.** Dataset distillation aims to learn a small synthetic dataset out of the original large dataset, such that the model trained on this small dataset can have a similar performance as the model trained on the original dataset. Wang et al. (2018) first introduced dataset distillation by gradient-based hyperparameter optimization. Bohdal et al. (2020) proposed a label distillation method to get soft labels for distilled images with the meta-learning framework. Zhao et al. (2021); Zhao & Bilen (2021b) proposed to match the gradient of the real dataset and the auxiliary dataset, and use data augmentation to boost the performance. Wang et al. (2022a); Zhao & Bilen (2021a) considered matching the distribution based on neural networks and largely reduced the memory cost. Matching Training Trajectories (Cazenavette et al., 2022) and its variants (Cazenavette et al., 2022; Liu et al., 2022; Du et al., 2022) are proposed to optimize the synthetic data by minimizing the distance between the synthetically trained parameters and the actual model training trajectories. Recent advances further push this direction: DATM (Guo et al., 2024) achieves lossless distillation by dynamically aligning trajectory difficulty to synthetic dataset size, and SeqMatch (Du et al., 2023) addresses accumulated trajectory error for more stable distillation.

**Federated Learning with Dataset Distillation.** Several recent works have tried to combine federated learning with dataset distillation. Specifically, Zhou et al. (2020) and Song et al. (2022) proposed a one-shot federated learning framework that allows all clients to distill their own dataset by local training and send the synthetic dataset to the server. The server will then use the collected distilled dataset for centralized training. Xiong et al. (2022) proposes a multi-shot framework that clients will distill their dataset for every communication round. Note that the above-mentioned works require (distilled) data transmission between clients and server: clients need to send their local distilled data to the server. Yet this incurs extra communication costs and data privacy concerns. Beyond reducing data transmission, a more fundamental question is how to effectively extract and utilize global information within the federated framework without any data sharing. FMKE (Liu et al., 2023) addresses this by condensing client knowledge into meta representations, though transmission of these representations is still required. DynaFed (Pi et al., 2023) takes a different

angle by leveraging global model dynamics to synthesize data on the server side, yet the synthesized data is used to directly augment server-side training, rather than to guide local client updates.

## 3 Preliminaries on Federated Learning

In this section, we will briefly introduce the basics of federated learning. Consider a federated system with $N$ participating clients, the goal of federated learning is to optimize the following objective function:

$$\min_{\mathbf{w} \in \mathbb{R}^d} \sum_{i=1}^{N} \frac{m_i}{M} \mathcal{L}(\mathbf{w}; \mathcal{D}_i), \tag{3.1}$$

where $\mathcal{L}$ denotes the loss function such as Cross Entropy loss, $\mathcal{D}_i$ denotes the training data on client $i$. Let $m_i = |\mathcal{D}_i|$ be the cardinality of $\mathcal{D}_i$ and we have $M = \sum_{i=1}^{N} m_i$ as total size of the training dataset.

**FedAvg.** FedAvg (McMahan et al., 2017) is designed to solve Eq (3.1) by coordinating multiple devices via a central server. Specifically, in each round of FedAvg, the server will broadcast the current global model $\mathbf{w}^t$ to the clients. Each client will locally optimize the corresponding component function in Eq (3.1) and obtain the local model $\mathbf{w}_i^{t+1}$, i.e., $\mathbf{w}_i^{t+1} = \operatorname{argmin}_{\mathbf{w}} \mathcal{L}(\mathbf{w}; \mathcal{D}_i)$, through multiple steps of stochastic gradient descent. The server will then aggregate (weighted average) the local models $\{\mathbf{w}_i^{t+1}\}_{i=1}^{N}$ to obtain the new global model.

**FedProx.** FedProx (Li et al., 2020) is one of the representative works which is designed to tackle both the data heterogeneity issue in federated learning and allows flexible performance on each local client. Specifically, it allows the local objectives to be solved inexactly such that the amount of local computation/communication can be adapted for different clients with different amounts of resources. To formally define this inexactness, Li et al. (2020) introduced the following definition:

**Definition 3.1** ($\gamma$-Inexactness). Given a function $h(\mathbf{w}, \mathbf{w}_0) = \mathcal{L}(\mathbf{w}) + \frac{\lambda}{2}\|\mathbf{w} - \mathbf{w}_0\|_2^2$ and $\gamma \in [0, 1]$, $\mathbf{w}^*$ is a $\gamma$-inexact solution of $\min_{\mathbf{w}} h(\mathbf{w}, \mathbf{w}_0)$ if $\|\nabla h(\mathbf{w}^*, \mathbf{w}_0)\|_2 \leq \gamma \|\nabla h(\mathbf{w}_0, \mathbf{w}_0)\|_2$.

Additionally, FedProx introduced a proximal term to the local objective to limit the local updates to be not too far away from its initial global model $\mathbf{w}^t$. In particular, the local objectives of FedProx become:

$$\mathbf{w}_i^{t+1} = \operatorname*{argmin}_{\mathbf{w}} \mathcal{L}(\mathbf{w}; \mathcal{D}_i) + \frac{\lambda}{2}\|\mathbf{w} - \mathbf{w}^t\|_2^2, \tag{3.2}$$

where $\lambda$ is the hyperparameter that controls the strength of the proximal term. We can observe that FedProx handles data heterogeneity by restricting the local updates to stay close to the initial global model.

## 4 Proposed Method

In this section, we introduce our proposed federated learning framework with Projected Trajectory Regularization (FedPTR) for tackling the data heterogeneity issue in federated learning. Due to the privacy-preserving design of federated learning, each client can only access their own training data. Therefore, the global model training trajectories contain all the information each client could possibly access with regard to other clients' local training data. Naturally, most existing works seek help from global models or global variables to mitigate the data heterogeneity issue. Different from existing works which mainly focus on utilizing only the last-step global model or the linear combination of past global models, FedPTR aims to provide a better way to exploit the global information hidden in the model training trajectories via Matching Training Trajectories (MTT) (Cazenavette et al., 2022). More specifically, we aim to extract an auxiliary dataset from the recent model training trajectories that could mimic the learning dynamics of the recent model update. Thus we can utilize such an auxiliary dataset for projecting the next-step training trajectory and provide better guidance to the local model training.

**Extract Global Information via MTT.** We found MTT (Cazenavette et al., 2022) to be a perfect tool for our global information extraction purposes. The general goal of MTT is to learn an informative but

---

**Algorithm 1** Federated Learning with Projected Trajectory Regularization (FedPTR)

---

1: **Input:** $T$, $m$, $\eta$, $\lambda$, $\beta$.
2: Initialize $\widetilde{\mathcal{D}}_i$ for all clients
3: **for** $t = 0$ **to** $T - 1$ **do**
4:    **for** each client $i \in [N]$ **do**
5:       Client $i$ receives $\mathbf{w}^t$ from the server
6:       $\widetilde{\mathbf{w}}_i^{t+1} = \mathbf{w}^t$
7:       **if** $t > m$ **then**
8:          $\widetilde{\mathcal{D}}_i \leftarrow \mathrm{MTT}(\mathbf{w}^{t-m}, \mathbf{w}^t, \widetilde{\mathcal{D}}_i, \beta)$,
9:          **for** $k = 0$ **to** $K - 1$ **do**
10:            $\widetilde{\mathbf{w}}_i^{t+1} \leftarrow \widetilde{\mathbf{w}}_i^{t+1} - \eta\nabla\mathcal{L}(\widetilde{\mathbf{w}}_i^{t+1}; \widetilde{\mathcal{D}}_i)$
11:          **end for**
12:          Set $\widetilde{\lambda} = \lambda$
13:       **else**
14:          Set $\widetilde{\lambda} = 0$
15:       **end if**
16:       Solve $\mathbf{w}_i^{t+1}$ as a $\gamma$-inexactness minimizer of: $\mathbf{w}_i^{t+1} \approx \underset{\mathbf{w}}{\arg\min} \, \mathcal{L}(\mathbf{w}; \mathcal{D}_i) + \frac{\widetilde{\lambda}}{2} \cdot \|\mathbf{w} - \widetilde{\mathbf{w}}_i^{t+1}\|_2^2$
17:       Client sends $\mathbf{w}_i^{t+1}$ to server
18:    **end for**
19:    Server aggregates $\{\mathbf{w}_i^{t+1}\}_{i=1}^N$ to get $\mathbf{w}^{t+1}$
20: **end for**

---

small synthetic dataset that approximates the true learning trajectories. Moreover, it only requires model parameters from previous training epochs which naturally fit the setting of federated learning. We summarize the detailed MTT process in Algorithm 2.

Specifically, we treat the last model in the actual training trajectories (denoted by $\mathbf{w}^{\mathrm{end}}$) as the teacher model in MTT and build a student model $\widehat{\mathbf{w}}$, which is obtained by multiple steps of gradient descent training on the synthetic dataset $\widetilde{\mathcal{D}}$ starting from $\mathbf{w}^{\mathrm{start}}$ in the trajectory. By requiring the student model to stay close to the teacher model, we force MTT to learn a synthetic dataset $\widetilde{\mathcal{D}}$ to that well-represents the recent learning dynamics. In detail, we have the following MTT loss:

$$\mathcal{L}_{\mathrm{MTT}}(\widetilde{\mathcal{D}}) = \|\widehat{\mathbf{w}} - \mathbf{w}^{\mathrm{start}}\|_2^2 / \|\mathbf{w}^{\mathrm{end}} - \mathbf{w}^{\mathrm{start}}\|_2^2, \tag{4.1}$$

where the normalization term ensures that the MTT loss remains effective even when the model's changes are small in the actual training trajectory. We can optimize $\mathcal{L}_{\mathrm{MTT}}(\widetilde{\mathcal{D}})$ over $\widetilde{\mathcal{D}}$ via gradient descent to obtain the auxiliary synthetic dataset.

**Projected Trajectory Regularization.** Once we obtain the auxiliary dataset via MTT, the following step is to project the next-step training trajectory and design our federated learning framework. We summarize our design at Algorithm 1 and give more details in the following [1].

Specifically, for each client (after certain training rounds $m$), we utilize MTT for extracting the auxiliary dataset $\widetilde{\mathcal{D}}_i$. Since the auxiliary dataset is designed to mimic the training trajectory from the $(t - m)$-th round global model to the $t$-th round global model, we can obtain our projected next-step model training trajectory $\widetilde{\mathbf{w}}_i^{t+1}$ by further performing several steps of gradient descent training on $\widetilde{\mathcal{D}}_i$ with learning rate $\eta$.

Inspired by FedProx, we also design a proximal term for guiding the local training. Different from FedProx, we don't use the last-step global model $\mathbf{w}^t$, instead, we hope the local update to stay close to our projected training trajectory $\widetilde{\mathbf{w}}_i^{t+1}$:

$$\min_{\mathbf{w}} \mathcal{L}(\mathbf{w}; \mathcal{D}_i) + \frac{\lambda}{2}\|\mathbf{w} - \widetilde{\mathbf{w}}_i^{t+1}\|_2^2, \tag{4.2}$$

---

[1] Algorithm 1 focused on the full participation setting where all clients participate in each round of federated learning. For partial participation cases, we can simply allow selected clients to perform MTT from the past trajectories it observed.

where $\lambda$ controls the strength of the proximal term. Through this design, we utilize the extracted global information from past training trajectories to help guide the local model update.

To make our algorithm more practical and provide more flexibility for local clients, we also follow FedProx and only require a $\gamma$-inexactness solution during local training. After finishing the local computation, clients will send their updated model to the server and the rest steps follow the same procedure as vanilla FedAvg.

---

**Algorithm 2** Matching Training Trajectories (MTT)

---
1: **Input:** $\mathbf{w}^{\mathrm{start}}$, $\mathbf{w}^{\mathrm{end}}$, $\widetilde{\mathcal{D}}_i$, $\beta$.
2: **for** $h = 0$ **to** $H - 1$ **do**
3:     $\widehat{\mathbf{w}} = \mathbf{w}^{\mathrm{start}}$
4:     **for** $r = 0$ **to** $R - 1$ **do**
5:        $\widehat{\mathbf{w}} \leftarrow \widehat{\mathbf{w}} - \beta \nabla \mathcal{L}(\widehat{\mathbf{w}}; \widetilde{\mathcal{D}}_i)$
6:     **end for**
7:     Computes $\mathcal{L}_{\mathrm{MTT}}(\widetilde{\mathcal{D}}_i) = \|\widehat{\mathbf{w}} - \mathbf{w}^{\mathrm{end}}\|_2^2 / \|\mathbf{w}^{\mathrm{end}} - \mathbf{w}^{\mathrm{start}}\|_2^2$
8:     Updates $\widetilde{\mathcal{D}}_i$ with respect to $\mathcal{L}_{\mathrm{MTT}}(\widetilde{\mathcal{D}}_i)$
9: **end for**
10: **Output:** $\widetilde{\mathcal{D}}_i$

---

**Initialization of the Auxiliary Dataset.** Previous studies (Cazenavette et al., 2022) have shown that using real data samples from true classes can improve the performance of MTT. However, it is not very practical in federated learning with heterogeneous data distributions, as it is nearly impossible for individual clients to possess real data samples from each class. Hence, we use an alternative strategy to initialize the auxiliary dataset $\widetilde{\mathcal{D}}_i$. Specifically, for those classes that have corresponding samples in the local's training data, we directly random sample (10-clients federated learning setuwith replacement) the desired amount of real data from the local training data. For those classes that do not have samples in the local training data, we adopt the random initialization strategy.

**Adaptive Regularization Parameter.** In Eq (3.2), we use a standard squared $L_2$ norm regularization as the proximal term. Following common machine learning practices, we usually adopt a constant $\lambda$ as the regularization parameter. However, one thing we notice is that in practice, when applying this $L_2$ norm regularization in neural network models, the model difference $\mathbf{w} - \widetilde{\mathbf{w}}_i^{t+1}$ can vary drastically among different convolutional layers. In Figure 1, we plot the $L_2$ norms of the four different convolutional layers in ConvNet(Gidaris & Komodakis, 2018), with the model trained under a 10-client federated learning setup, and each client's training data sampled from the original CIFAR-10 dataset following Dirichlet distribution ($\alpha = 0.01$).

From Figure 1, we observe that the $L_2$ norm of $\mathbf{w} - \widetilde{\mathbf{w}}_i^{t+1}$ between layers varies and it also changes along with different phases of training. For example, at the beginning of training, the fourth layer has the largest $L_2$ norm while the second layer has the largest $L_2$ norm when the training is about to end. Such a phenomenon indicates that it might be helpful to use a layer-adaptive $\lambda$ rather than a constant.

Specifically, let $\mathbf{w}_{[j]}$ denote parameters for the $j$-th layer in $\mathbf{w}$ and $\widetilde{\mathbf{w}}_{i,[j]}^{t+1}$ denotes parameters for the $j$-th layer in $\widetilde{\mathbf{w}}$, we can define the adaptive $\lambda_j$ as follows:

$$\lambda_j = \frac{1}{\left\|\mathbf{w}_{[j]} - \widetilde{\mathbf{w}}_{i,[j]}^{t+1}\right\|_2} \cdot \lambda. \tag{4.3}$$

Here, we compare our layer adaptive $\lambda$ with the standard fixed $\lambda$. We report results on CIFAR-10 and CIFAR-100 with the parameter of Dirichlet distribution $\alpha = 0.01$ and $0.04$ as shown in Table 1. We can observe that our FedPTR adopting layer adaptive $\lambda$ achieves better performances compared with the fixed $\lambda$ setting. By default, we report all our experimental results in Section 6 with the layer adaptive $\lambda$.

Table 1: Comparisons of layer-adaptive $\lambda$ with fixed $\lambda$ in FedPTR on CIFAR-10 and CIFAR-100 dataset.

| Reg. param. | Dirichlet $\alpha = 0.01$ | | Dirichlet $\alpha = 0.04$ | |
|---|---|---|---|---|
| | CIFAR 10 | CIFAR 100 | CIFAR 10 | CIFAR 100 |
| Fixed $\lambda$ | 62.10 | 41.26 | 67.68 | 42.41 |
| Layer-adaptive $\lambda$ | 69.04 | 44.98 | 72.78 | 45.87 |

**Server-Side FedPTR.** One problem of FedPTR presented in Algorithm 1 is that each client needs to perform Matching Training Trajectories and update their own auxiliary dataset in each communication round, which would introduce extra computation costs. To make our method smoothly accommodate resource-constrained devices (e.g., mobile phones, robots, drone swarms), we also provide a server-side FedPTR algorithm (denoted by FedPTR-S), which only optimizes the auxiliary dataset through MTT in the global server. We summarize the detailed algorithm in Algorithm 3. Specifically, in each communication round, the global server first starts to match training trajectories and optimize a unified $\widetilde{\mathbf{w}}^{t+1}$. Then, each client will receive both $\mathbf{w}^t$ and $\widetilde{\mathbf{w}}^{t+1}$ for local training. Note that different from client-side FedPTR as shown in Algorithm 1, the server-side version initializes the auxiliary dataset using random noise without any real data from local clients (see a detailed comparison of different initialization strategies in B.1). Therefore, our FedPTR-S can preserve data privacy while also making participating devices free from extra computation burdens.

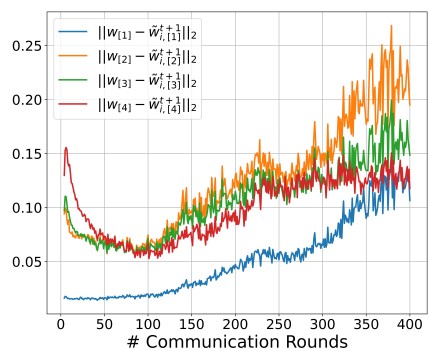

Figure 1: Comparisons of layer-wise $L_2$ norm $\left\|\mathbf{w}_{[j]} - \widetilde{\mathbf{w}}_{i,[j]}^{t+1}\right\|_2$ against the communication rounds, where $j$ denotes $j$-th layer. The layer-wise $L_2$ norm values vary significantly with different layers and communication rounds.

## 5 Theoretical Analysis

In this section, we provide the theoretical convergence analysis for the proposed FedPTR framework. We first state several assumptions needed for the convergence analysis.

**Assumption 5.1.** The loss function on the $i$-th client $\mathcal{L}(\mathbf{w}; \mathcal{D}_i)$ is $L$-smooth, i.e., $\forall \mathbf{w}, \mathbf{v} \in \mathbb{R}^d$,

$$\left|\mathcal{L}(\mathbf{w}; \mathcal{D}_i) - \mathcal{L}(\mathbf{v}; \mathcal{D}_i) - \langle \nabla \mathcal{L}(\mathbf{v}; \mathcal{D}_i), \mathbf{w} - \mathbf{v} \rangle\right| \leq \frac{L}{2} \|\mathbf{w} - \mathbf{v}\|_2^2.$$

**Assumption 5.2.** The gradient of the local loss function on the $i$-th client $\nabla \mathcal{L}(\mathbf{w}; \mathcal{D}_i)$ is $B$-locally dissimilar, i.e.,

$$\frac{1}{N} \sum_{i \in [N]} \|\nabla \mathcal{L}(\mathbf{w}; \mathcal{D}_i)\|_2^2 \leq B^2 \|\nabla \mathcal{L}(\mathbf{w})\|_2^2.$$

Assumption 5.1 and 5.2 are standard assumptions in federated learning optimization problems (Acar et al., 2021; Li et al., 2020; Karimireddy et al., 2020b; Xu et al., 2021; Gao et al., 2022). Assumption 5.2 describes the dissimilarity between the gradient of local objectives and the global objective. Note that the commonly used bounded variance assumption (Li et al., 2020; Reddi et al., 2020; Wang et al., 2022b), $\frac{1}{N} \sum_{i \in [N]} \|\nabla \mathcal{L}(\mathbf{w}; \mathcal{D}_i) - \nabla \mathcal{L}(\mathbf{w})\|^2 \leq \sigma^2$, is another standard way to characterize client heterogeneity, and is often used as an alternative to Assumption 5.2.

**Assumption 5.3.** The auxiliary loss function on the $i$-th client $\widetilde{\mathcal{L}}(\mathbf{w}; \widetilde{\mathcal{D}}_i)$ has $\ell_2$-bounded gradient dissimilarity with global objective function $\mathcal{L}(\mathbf{w})$, i.e., for all $\mathbf{w} \in \mathbb{R}^d$, taken over the randomness in the auxiliary dataset, we have

$$\mathbb{E}_{\widetilde{\mathcal{D}}_i}[\|\nabla \mathcal{L}(\mathbf{w}) - \nabla \widetilde{\mathcal{L}}(\mathbf{w}; \widetilde{\mathcal{D}}_i)\|_2^2] \leq \sigma_d^2.$$

Assumption 5.3 measures the gradient dissimilarity between the global loss function and the training loss on the auxiliary dataset $\widetilde{\mathcal{D}}_i$, which implies that the MTT algorithm is actually well-functioning: the extracted auxiliary dataset indeed mimics the actual global model training trajectories.

In the following, we will state the theoretical convergence analysis for our proposed FedPTR. For expository purposes, we assume $K = 1$, i.e., performing 1 step of auxiliary model update in each round, though our convergence analysis can also be extended to cases when $K > 1$.

**Theorem 5.4** (Non-convex FedPTR convergence). Under Assumptions 5.1-5.3, assume the local objective functions $\mathcal{L}(\mathbf{w}; \mathcal{D}_i)$ are non-convex, and there exists $L_- > 0$, such that $\nabla^2 \mathcal{L}(\mathbf{w}; \mathcal{D}_i) \succeq -L_- \mathbf{I}$ with $\mu = \lambda - L_- > 0$. If $\eta \leq \frac{1}{2\lambda}$, $\gamma$ is selected to be sufficiently small, $\mu$ and $\lambda$ satisfies

$$\frac{1}{6\lambda} - \frac{4L^2 B^2}{\lambda \mu^2} - \frac{2L^2}{\lambda \mu^2} - \frac{1}{\lambda N} - \frac{2LB^2}{\mu^2} - \frac{4L}{\mu^2} \geq 0, \tag{5.1}$$

then the iterations of FedPTR satisfy

$$\min_{t \in [T]} \mathbb{E}[\|\nabla \mathcal{L}(\mathbf{w}_t)\|_2^2] \leq \mathcal{O}\left( \frac{\mathbb{E}[\mathcal{L}(\mathbf{w}_0)] - \mathbb{E}[\mathcal{L}(\mathbf{w}_T)]}{6\lambda T} \right) + \mathcal{O}(\eta^2 \sigma_d^2). \tag{5.2}$$

Theorem 5.4 suggests that the gradient norm of FedPTR has an upper bound that contains two terms. The first term vanishes as the communication round $T$ increases and the second term is related to $\sigma_d^2$, i.e., the gradient dissimilarity between global training loss and the training loss on the auxiliary dataset. This implies that if the auxiliary gradient differs significantly from the global gradient, it will hurt FedPTR's performance as larger $\sigma_d$ leads to worse convergence. In such a case, the auxiliary gradient cannot represent the global gradient direction well, hence to obtain a desired convergence performance, it requires a smaller $\eta$ to control the auxiliary model $\widetilde{\mathbf{w}}_{t+1}$ to stay close to the global initial model $\mathbf{w}_t$ and obtain smaller second term in Eq (5.2).

**Remark 5.5.** Theorem 5.4 can be extended to our proposed server-side FedPTR (Algorithm 3) with slightly modifying Assumption 5.3 to $\|\nabla \mathcal{L}(\mathbf{w}) - \nabla \widetilde{\mathcal{L}}(\mathbf{w}; \widetilde{\mathcal{D}})\|_2^2 \leq \sigma_d^2$, which describes the dissimilarity between the server-side auxiliary loss and the global objective function.

**Corollary 5.6.** If choosing the auxiliary learning rate as $\eta = \Theta\left(\frac{1}{\sqrt{T}}\right)$, then the convergence rate for Algorithm 1 satisfies

$$\min_{t \in [T]} \mathbb{E}[\|\nabla \mathcal{L}(\mathbf{w}^t)\|_2^2] = \mathcal{O}\left(\frac{1}{T}\right). \tag{5.3}$$

Corollary 5.6 suggests that with an appropriately chosen learning rate, FedPTR achieves a convergence rate of $\mathcal{O}(1/T)$ in stochastic non-convex optimization settings, which matches the convergence rate of federated learning with proximal term such as FedProx (Li et al., 2020) and FedDyn (Acar et al., 2021).

**Remark 5.7.** We note that the rate in Corollary 5.6 is established under the $\gamma$-inexact local-solver abstraction. Rather than explicitly tracking mini-batch gradient noise, our analysis assumes that each client returns a $\gamma$-inexact solution of its regularized local subproblem. Thus, stochastic local training procedures are covered insofar as their returned local models satisfy the $\gamma$-inexactness condition, and Theorem 5.4 and Corollary 5.6 should not be interpreted as variance-explicit stochastic-gradient convergence results.

## 6 Experiments

In this section, we present comprehensive experimental results to show the effectiveness of FedPTR when competing with other state-of-the-art federated learning methods under various heterogeneous data partitions. We first introduce our experimental settings in 6.1 and give a detailed analysis of our experimental results and ablation study in the following sections. We report the average accuracy of the last five global rounds on the test set.

### 6.1 Experimental Settings

**Datasets.** We conduct experiments on four benchmark datasets: FashionMNIST (Xiao et al., 2017), CIFAR-10 (Krizhevsky et al., 2009), CIFAR-100 (Krizhevsky et al., 2009), and TinyImageNet (Le & Yang, 2015).

Table 2: Comparisons of test accuracy for FedPTR (FedPTR-S) and other baselines in training different datasets and under different participation ratios with 10 clients. **Bold** and underline represent the best and second best results.

| Participation Ratio | Dataset | FedAvg | FedProx | Scaffold | FedDyn | FedDC | FedPTR | FedPTR-S |
|---|---|---|---|---|---|---|---|---|
| | | | | **Dirichlet(0.01)** | | | | |
| | FMNIST | 83.69 | 83.75 | 82.21 | 84.36 | 78.23 | 85.11 | **85.48** |
| | CIFAR-10 | 55.43 | 55.94 | 56.58 | 62.29 | 52.55 | **69.04** | 68.16 |
| | CIFAR-100 | 37.32 | 37.33 | 42.02 | 41.73 | 39.52 | **44.98** | 44.82 |
| 100% | | | | **Dirichlet(0.04)** | | | | |
| | FMNIST | 85.05 | 84.94 | 84.59 | 85.26 | 82.17 | **86.80** | 85.90 |
| | CIFAR-10 | 65.44 | 65.46 | 68.98 | 70.26 | 68.93 | 72.78 | **73.07** |
| | CIFAR-100 | 39.65 | 39.63 | 44.66 | 45.57 | 44.23 | 45.87 | **46.10** |
| | | | | **Dirichlet(0.01)** | | | | |
| | FMNIST | 81.02 | 81.05 | 75.57 | 79.39 | 73.18 | **81.76** | 81.71 |
| | CIFAR-10 | 49.90 | 50.28 | 54.91 | 56.81 | 50.65 | **56.95** | 54.24 |
| | CIFAR-100 | 37.07 | 37.04 | 37.35 | 37.61 | 37.98 | **40.11** | 39.49 |
| 50% | | | | **Dirichlet(0.04)** | | | | |
| | FMNIST | 81.75 | 81.73 | 77.42 | 80.71 | 77.44 | 82.85 | **83.05** |
| | CIFAR-10 | 56.84 | 56.80 | 57.58 | 59.45 | 61.31 | **65.60** | 65.06 |
| | CIFAR-100 | 38.22 | 38.24 | 38.33 | 40.68 | 40.50 | 43.63 | **44.99** |
| | | | | **Dirichlet(0.01)** | | | | |
| | FMNIST | 75.89 | 76.40 | 73.05 | 75.09 | 73.09 | **76.79** | 76.78 |
| | CIFAR-10 | 41.82 | 43.06 | 47.92 | 45.20 | 44.01 | **48.94** | 47.41 |
| | CIFAR-100 | 29.68 | 29.65 | 29.18 | 30.80 | 32.67 | **35.41** | 33.64 |
| 30% | | | | **Dirichlet(0.04)** | | | | |
| | FMNIST | 77.79 | 78.75 | 77.11 | 78.89 | 76.99 | 79.03 | **79.48** |
| | CIFAR-10 | 48.50 | 48.52 | 53.14 | 44.21 | 55.72 | 56.67 | **57.75** |
| | CIFAR-100 | 32.95 | 32.87 | 37.46 | 36.82 | 37.57 | 37.49 | **38.21** |

Table 3: Comparisons of FedPTR (FedPTR-S) with other baselines for partial participation in training different datasets with 40 clients. **Bold** and underline represent the best and second best results.

| Setting | Dataset | FedAvg | FedProx | Scaffold | FedDyn | FedDC | FedPTR | FedPTR-S |
|---|---|---|---|---|---|---|---|---|
| | | | | **Dirichlet(0.01)** | | | | |
| | CIFAR-100 | 32.68 | 32.71 | 36.07 | 29.91 | 35.66 | **37.50** | 36.40 |
| 40 clients with | | | | **Dirichlet(0.04)** | | | | |
| **25%** participation ratio | FMNIST | 83.73 | 83.77 | 79.76 | 78.78 | 82.71 | **85.92** | 84.68 |
| | CIFAR-10 | 58.00 | 58.06 | 58.32 | 54.11 | 55.13 | **60.30** | 59.73 |
| | CIFAR-100 | 38.35 | 38.32 | 40.34 | 38.67 | **42.74** | 41.90 | 41.11 |

We follow the commonly used mechanism to perform data partition through Dirichlet distribution (Wang et al., 2020a). The parameter $\alpha$ used in Dirichlet sampling determines the non-i.i.d. degree. With smaller $\alpha$, non-i.i.d. degree becomes higher. In our experiments, we focus on the high-level data heterogeneity with $\alpha = 0.01$ and $\alpha = 0.04$.

**Baselines.** We compare our method FedPTR (and FedPTR-S) with five state-of-the-art federated learning methods: FedAvg (McMahan et al., 2017), FedProx (Li et al., 2020), SCAFFOLD (Karimireddy et al., 2020b), FedDyn (Acar et al., 2021), and FedDC (Gao et al., 2022). All the baseline methods except FedAvg use global information especially on the consistency of the global model and local model while FedPTR integrates the global information by training an auxiliary dataset based on the global parameters.

**Hyper-parameters and Model Architectures.** In our experiments, we set the batch size as 500 and the local training epoch as 1. By default, we use SGD with a learning rate $\eta = 0.01$ and momentum of 0.5 as the optimizer. We use 10 images per class for FashionMNIST/CIFAR-10 and 2 images per class for CIFAR-100. We set trajectory projection steps $K = 5$ and the base $\lambda$ for layer-adaptive regularization parameter as $\lambda = 0.05$. We also tune the hyperparameters in all other baselines for their best performances. In matching

training trajectories part (Algorithm 2), following (Cazenavette et al., 2022), we utilize a learnable $\beta$ with initialized $\beta = 0.01$, trajectory matching outer steps $H = 20$, inner steps $R = 10$, while the rest of the MTT hyper-parameters are set as the default values from (Cazenavette et al., 2022). In terms of the model architecture, we follow (Cazenavette et al., 2022) and adopt ConvNet (Gidaris & Komodakis, 2018) as the default CNN architecture.

## 6.2 Comparison of Different Federated Learning Methods on Heterogeneous Data

We compare our method FedPTR (and FedPTR-S) with five state-of-the-art federated learning optimization methods: FedAvg (McMahan et al., 2017), FedProx (Li et al., 2020), SCAFFOLD (Karimireddy et al., 2020b), FedDyn (Acar et al., 2021), and FedDC (Gao et al., 2022), on varying data heterogeneity, the number of clients, and participation rate. Table 2 shows the comparison results of our method FedPTR with baselines for the scenario of full participation and partial participation with 10 clients. We perform evaluations on two different heterogeneous data partitions, Dirichlet $\alpha = 0.04$ and Dirichlet $\alpha = 0.01$. In the full participation setting (participation ratio 100%), we observe that our method outperforms previous methods regarding test accuracy on all datasets. In particular, we achieve the absolute improvement of 6.75% compared with the best-performing baseline FedDyn and over 12% improvement over other baselines on CIFAR-10 dataset with Dirichlet $\alpha = 0.01$. Moreover, it can be seen that previous methods suffer from performance decreases of up to 16.38% when the data partition becomes more heterogeneous (i.e., from Dirichlet $\alpha = 0.04$ to Dirichlet $\alpha = 0.01$), while our method only reduces the test accuracy by up to 4.91%. Such results demonstrate that our method is consistently robust against extremely heterogeneous training data. In partial participation, we evaluate our method and baselines with 50% and 30% participation ratios. We can see that our method obtains relative improvements of 0.9% ∼ 17.7% and 1.2% ∼ 19.3% on 50% participation and 30% participation scenarios compared with FedAvg. These results clearly show the effectiveness of our method for the scenario of partial participation. To further verify our method in the scenario of more clients, we additionally compare FedPTR with baselines in 40 clients and 25% participation ratio as shown in Table 3. We observe that our method FedPTR (and FedPTR-S) still maintains excellent performance compared with the previous methods. Note that in the setting of Dirichlet $\alpha = 0.01$, we only evaluate our method on CIFAR-100 since FashionMNIST and CIFAR-10 with fewer categories cannot make each client have training data.

## 6.3 Reducing the Updates of Auxiliary Dataset

The original design of FedPTR (FedPTR-S) requires the clients/server to update the auxiliary datasets in each communication round. In this section, we explore how performance scales if we reduce the frequency of the auxiliary dataset updates. In Figure 2, we compare the test accuracy against the communication rounds on CIFAR-10 Dirichlet $\alpha = 0.01$ by performing one MTT update per 1, 4, 10, and 40 rounds. We additionally try a one-shot MTT strategy by only updating the auxiliary dataset once. Note that when there is no MTT update over the auxiliary dataset, we still need to perform the trajectory projection step (with the "old" auxiliary dataset) for local update regularization. We find that our proposed FedPTR method does not suffer much in performance even when the number of updates of the auxiliary dataset is reduced by 75%. Moreover, even the most efficient FedPTR with one-shot MTT can still outperform the other baseline methods such as FedDyn (Acar et al., 2021) on CIFAR-10.

## 6.4 Changing the Size of Auxiliary Dataset

We study how the size of the auxiliary dataset affects our final performances. For this, we conduct experiments comparing FedPTR using different sizes of the auxiliary dataset on CIFAR-10 and CIFAR-100 dataset and show the results in Table 4. For CIFAR-10, we explore 5 different sizes, and the results suggest that a moderate size of the auxiliary dataset leads to the best model training performances. When the size is below 100, the test accuracy is largely affected due to less representation power limited by the auxiliary dataset size. On the other hand, when the size grows beyond 100, there is a clear marginal effect and the influence of the size is not as obvious as before (may even lose performances due to insufficient optimization of the large auxiliary data). For CIFAR-100, a similar trend can also be observed. Note that using a larger

auxiliary dataset leads to a larger computational cost when performing MTT updates in each communication round. In practice, we would be picking a moderate auxiliary dataset size (such as 100) for the best of both performance and computational cost.

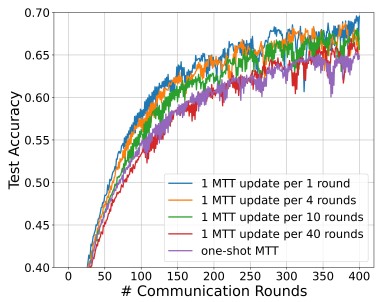

Figure 2: Comparisons of FedPTR test accuracy against communication rounds with the different frequency of MTT updates on CIFAR-10 dataset with Dirichlet $\alpha = 0.01$.

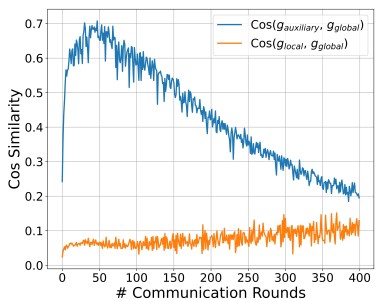

Figure 3: Comparisons of the cosine similarity between the auxiliary gradient and the global gradient, as well the cosine similarity between the local gradient and the global gradient.

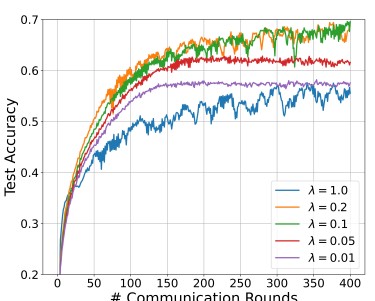

Figure 4: $\lambda$ sensitivity test of FedPTR on CIFAR-10 with Dirichlet $\alpha = 0.01$.

## 6.5 Quality of Extracted Global Information

In this subsection, we give a brief analysis on the quality of the extracted global information by the auxiliary dataset. To do so, we measure the similarity between the gradient of the training loss on the auxiliary dataset (denoted by the auxiliary gradient) and the global gradient. A high similarity between the two gradients could indicate that extracted auxiliary datasets well-approximate the recent learning dynamics and can well project the next-step training trajectory. For computational simplicity, here we directly use the global update direction on the server to approximate the global gradient direction, i.e., $\mathbf{w}^t - \mathbf{w}^{t+1}$. And we use $\mathbf{w}^t - \widetilde{\mathbf{w}}_i^{t+1}$ to approximate the auxiliary gradient. To further demonstrate the quality of extracted global information, we also compute the cosine similarity of local gradient $\mathbf{w}^t - \mathbf{w}_i^{t+1}$ and global gradient. Note that we compute two cosine similarities on vanilla FedAvg, i.e., clients keep the update of the auxiliary dataset and projected training trajectory while not using any regularization term to avoid the auxiliary

Table 4: Comparisons of FedPTR with different sizes of the auxiliary dataset with Dirichlet $\alpha = 0.01$.

| Dataset | Size of $\widetilde{\mathcal{D}}$ | Test Acc |
|---------|------------|----------|
| CIFAR-10 | 10 | 65.83 |
| | 50 | 67.02 |
| | 100 | 69.04 |
| | 150 | 68.46 |
| | 200 | 69.00 |
| CIFAR-100 | 100 | 44.57 |
| | 200 | 44.98 |
| | 300 | 44.67 |

dataset influencing the training gradient. We present our results for one randomly chosen client in training a CIFAR-100 model with Dirichlet $\alpha = 0.01$ in Figure 3. Compared with the local gradient, the auxiliary gradient has a much higher similarity to the global gradient, which suggests that FedPTR can indeed extract global information from the model training trajectory. Note that the cosine similarity of the auxiliary gradient and global gradient decreases with the increase of the communication rounds. This suggests that it is harder to extract global information at the later training phase. Nevertheless, we can still find that the auxiliary gradient shares greater similarity to the global gradient direction compared with the local gradient along the entire training trajectory.

## 6.6 Ablation Study

**Sensitivity Analysis of Regularization weight.** One of the important hyperparameters of FedPTR is the weight of the regularization term $\lambda$. To test its sensitivity, we consider the non-i.i.d. setting of full participation on CIFAR-10 with Dirichlet $\alpha = 0.01$. We conduct the sensitivity test from the candidate

set $\{0.01, 0.05, 0.1, 0.2, 1.0\}$. Figure 4 shows the convergence plot at varying $\lambda$, while we keep all other hyperparameters of FedPTR consistent. We can observe that the best test accuracy is obtained when $\lambda = 0.1$. The small $\lambda$ (e.g., 0.01) may not have any effect, and the large $\lambda$ (e.g., 1.0) possibly force $\mathbf{w}$ to be close to $\widetilde{\mathbf{w}}_i^K$ thus causing slow convergence.

Table 5: Comparisons of FedPTR-S with different trajectory matching steps with Dirichlet $\alpha = 0.01$ and 0.04.

| | Dirichlet $\alpha$ | Trajectory matching steps $H$ | | | |
|---|---|---|---|---|---|
| | | 5 | 10 | 20 | 30 |
| CIFAR-10 | $\alpha = 0.01$ | 67.00 | 68.12 | 68.16 | 67.04 |
| | $\alpha = 0.04$ | 72.85 | 71.44 | 73.07 | 73.20 |
| CIFAR-100 | $\alpha = 0.01$ | 44.56 | 44.53 | 44.82 | 44.89 |
| | $\alpha = 0.04$ | 45.60 | 45.89 | 46.10 | 45.85 |

**Impact of trajectory matching steps.** In Table 5, we present the performance of our method with varying total trajectory matching steps $H$ used in Algorithm 2. Specifically, we consider 10 clients, full participation settings with Dirichlet $\alpha = 0.01$ and Dirichlet $\alpha = 0.04$. We can see that our method is stable with respect to different trajectory matching steps. Even if we reduce the steps from 20 to 5, the test accuracy obtained on CIFAR-10 and CIFAR-100 only decreases by 1.16% and 0.5%, respectively.

**Impact of trajectory projecting steps.** To test the sensitivity of the trajectory projection steps $K$, we consider CIFAR-10 with Dirichlet 0.01. Figure 5 shows convergence plots for different $K$ configurations. We can observe that our method is relatively stable when $k = 3$ and $k = 5$. However, setting $k = 1$ would lead to slower convergence and a slight performance decrease, and setting $k = 10$ could accelerate the training in the early stage but result in instability and make it difficult to converge.

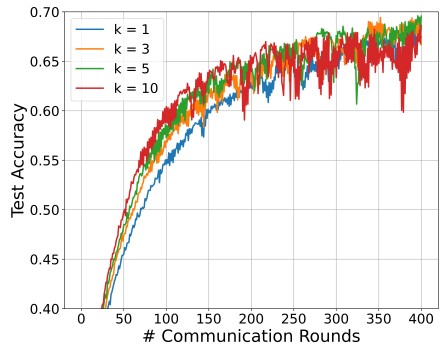

Figure 5: Comparisons of FedPTR with different trajectory projecting steps $K$ on CIFAR-10 with Dirichlet $\alpha = 0.01$.

## 7 Privacy leakage of FedPTR

For privacy leakage analysis of our proposed method, we argue that although FedPTR does not incorporate explicit privacy protection mechanisms, it does not introduce additional privacy vulnerabilities compared to existing federated learning approaches such as FedAvg (McMahan et al., 2017) and FedDyn (Acar et al., 2021). This is because all client-server communications in FedPTR consist solely of model weights and updates, identical to the communication patterns in FedAvg (McMahan et al., 2017) and FedDyn (Acar et al., 2021). To empirically validate that FedPTR does not incur additional privacy leakage, we conduct membership inference attacks (MIA) on the final models trained using different methods. We present MIA results for 10-client federated learning with $\alpha = 0.04$ on the CIFAR-100 dataset in Table 6, employing two well-established MIA techniques: (1) the loss-based method and (2) the Likelihood Ratio Attack (LiRA). The reported accuracy represents the success rate of the MIA method in correctly predicting whether a sample belongs to the original training set. Notably, MIA test accuracy closer to 50% (equivalent to random guessing) indicates minimal privacy leakage. Our results demonstrate that FedPTR achieves comparable privacy preservation levels to other baseline methods, confirming our theoretical analysis.

### 7.1 Inadequacy of Trivial Synthetic Data Integration

In this section, we present a naive approach that directly combines distilled datasets with local data and analyze its limitations. Since MTT (Cazenavette et al., 2022) was originally designed for distilling small

synthetic datasets, a straightforward solution for addressing data heterogeneity would be to apply MTT (Cazenavette et al., 2022) to generate synthetic data and combine it with available local data for FedProx (Li et al., 2020) training. However, this trivial approach proves inadequate for addressing data heterogeneity issues, particularly under highly non-i.i.d. data distributions across clients, as demonstrated in Table 7. The results show that simply augmenting FedProx training with synthetic data from MTT (Cazenavette et al., 2022) yields even worse performance than FedAvg. This degradation occurs because the synthetic data generated by MTT (Cazenavette et al., 2022) in federated settings may inadequately represent the original training data information. Unlike the original MTT (Cazenavette et al., 2022), which can leverage multiple training trajectories and restart strategies, federated settings impose realistic constraints that limit these capabilities. Consequently, direct training on such synthetic data can actually harm performance. In contrast, our approach fundamentally differs from this naive combination. We utilize the auxiliary data produced by MTT (Cazenavette et al., 2022) specifically to guide trajectory prediction for subsequent training steps—precisely the task MTT (Cazenavette et al., 2022) was designed for (i.e., matching recent training trajectories). This targeted usage represents a fundamental departure from simply combining existing methods like FedProx (Li et al., 2020) and MTT (Cazenavette et al., 2022).

Table 6: Membership inference attack for global model with $\alpha = 0.04$ on CIFAR-100 dataset.

|  | FedAvg | FedDyn | FedPTR |
| --- | --- | --- | --- |
| loss-based | 56.37 | 61.67 | 57.46 |
| LIRA | 61.06 | 68.29 | 65.59 |

Table 7: Comparisons of directly using auxiliary dataset and FedPTR on three datasets.

|  | FMNIST | CIFAR-10 | CIFAR-100 |
| --- | --- | --- | --- |
| Extra synthetic data | 76.23 | 50.08 | 27.78 |
| FedAvg | 83.69 | 55.45 | 37.32 |
| FedPTR | 85.11 | 69.04 | 44.98 |
| FedPTR-S | 85.48 | 68.16 | 44.82 |

## 8  Conclusion

Data heterogeneity issue is a critical factor for the practical deployment of Federated Learning algorithms. In this work, we propose a novel federated learning framework FedPTR that handles extreme data heterogeneity across local clients. The proposed method allows each client or the server to extract essential global information from the observed global model training trajectory and utilize it to project the next-step model trajectory for local training regularization. The exhaustive experimental results on various benchmark datasets and different non-i.i.d. settings clearly demonstrate the effectiveness of our proposed FedPTR in training real-world machine learning models. One potential limitation of FedPTR is that it cannot be directly applied to language tasks since the application of MTT on language tasks is still an open question. We will leave it as our future work.

## Acknowledgments

We thank the anonymous reviewers for their helpful comments. This work is partially supported by the National Science Foundation under Grant No. 2348541. The views and conclusions contained in this paper are those of the authors and should not be interpreted as representing any funding agencies.

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

# A  Server-Side FedPTR and Implementation Details

## A.1  Server-Side FedPTR

We present our Server-Side FedPTR in Algorithm 3, which only updates the auxiliary dataset through MTT in the global server. At the start of each communication round, the global server first matches training

trajectories and optimizes a unified $\widetilde{\mathbf{w}}^{t+1}$. Then, each client will receive both $\mathbf{w}^t$ and $\widetilde{\mathbf{w}}^{t+1}$ for local training. Different from client-side FedPTR, server-side FedPTR only uses random noise to initialize the auxiliary dataset. Therefore, our FedPTR-S maintains privacy-preserving while also making participating devices free from extra computation burdens.

---

**Algorithm 3** Server-Side FedPTR (FedPTR-S)

---

1: **Input:** $T$, $m$, $\eta$, $\lambda$.
2: Initialize $\widetilde{\mathcal{D}}$ for server
3: **for** $t = 0$ **to** $T - 1$ **do**
4:     Server starts to match training trajectories:
5:     **if** $t > m$ **then**
6:         $\widetilde{\mathcal{D}} \leftarrow \mathrm{MTT}(\mathbf{w}^{t-m}, \mathbf{w}^t, \widetilde{\mathcal{D}})$
7:         **for** $k = 0$ **to** $K - 1$ **do**
8:             $\widetilde{\mathbf{w}}^{t+1} \leftarrow \widetilde{\mathbf{w}}^{t+1} - \eta \nabla \mathcal{L}(\widetilde{\mathbf{w}}^{t+1}; \widetilde{\mathcal{D}})$
9:         **end for**
10:         Set $\widetilde{\lambda} = \lambda$
11:     **else**
12:         $\widetilde{\mathbf{w}}^{t+1} = \mathbf{w}^t$
13:         Set $\widetilde{\lambda} = 0$
14:     **end if**
15:     **for** each client $i \in [N]$ **do**
16:         Client $i$ receives $\mathbf{w}^t$ and $\widetilde{\mathbf{w}}^{t+1}$ from the server
17:         Solve $\mathbf{w}_i^{t+1}$ as a $\gamma$-inexactness minimizer of: $\mathbf{w}_i^{t+1} \approx \underset{\mathbf{w}}{\arg\min}\, \mathcal{L}(\mathbf{w}; \mathcal{D}_i) + \frac{\widetilde{\lambda}}{2} \cdot \|\mathbf{w} - \widetilde{\mathbf{w}}^{t+1}\|_2^2$
18:         Client sends $\mathbf{w}_i^{t+1}$ to server
19:     **end for**
20:     Server aggregates $\{\mathbf{w}_i^{t+1}\}_{i=1}^N$ to get $\mathbf{w}^{t+1}$
21: **end for**

---

### A.2 Detailed Description of Data Sampling

As we have mentioned in experimental settings, for all experiments, we adopt Dirichlet data sampling methods, which are commonly used in federated learning papers(Acar et al., 2021; Wang et al., 2020a). For Dirichlet sampling methods, each client $j$ is allocated a proportion of the samples from each label $k$ according to Dirichlet distribution. Specifically, we sample $p_k \sim Dir_N(\alpha)$ and allocate a $p_{kj}$ proportion of data samples of class $k$ to client $j$, where $Dir_N(\alpha)$ denotes the Dirichlet distribution and $\alpha > 0$ is a concentration parameter. A smaller $\alpha$ indicates a higher non-i.i.d. degree when sampling data, and we focused on the high-level data heterogeneity with $\alpha = 0.01$ and $\alpha = 0.04$. Table 8 shows a typical Dirichlet sampling result for different settings. We can observe that when choosing 40 clients on CIFAR-10, it can easily assign 0 data samples to certain clients due to low value and a small number of classes. That is the reason why we do not conclude experiments with more clients on CIFAR-10 and FashionMNIST.

## B Additional Experiments

### B.1 Initialization of the Auxiliary Dataset

We compare five strategies for initializing the auxiliary dataset on CIFAR-10 with Dirichlet $\alpha = 0.01$, including zero init (all-zero pixels), Gaussian init ($\mathcal{N}(0,1)$ per pixel), uniform init ($\mathcal{U}(-1,1)$ per pixel, testing sensitivity to the noise distribution), local-random init (each synthetic slot filled with a randomly sampled local real image while ignoring class labels, which isolates the contribution of class-image alignment from that of simply using real images), and our class-aware init (using local real data with matching class labels when available). As shown in Table 9, zero initialization clearly underperforms (65.12), confirming that MTT requires a non-trivial starting point. The two pure-noise inits (Gaussian 67.50, uniform 68.51)

Table 8: Dirichlet sampling result for different settings.

| Dataset | $\alpha$ | # clients | # samples in each client |
|---------|----------|-----------|--------------------------|
| FMNIST | 0.01 | 10 | 12032, 4244, 6003, 6037, 6002, 6165, 5963, 5760, 2602, 5192 |
| CIFAR-10 | 0.01 | 10 | 2972, 9419, 2936, 5779, 4111, 5003, 4949, 4979, 5615, 4237 |
| CIFAR-100 | 0.01 | 10 | 2690, 5209, 4764, 6463, 4467, 5679, 3983, 5910, 5497, 5338 |
| CIFAR-100 | 0.01 | 40 | 735, 1371, 1365, 761, 1123, 2194, 2399, 531, 848, 1549, 2378, 569, 2007, 984, 518, 608, 1250, 780, 2601, 2784, 1408, 1831, 1108, 2166, 523, 814, 1281, 579, 1067, 1172, 803, 1085, 902, 2603, 1237, 940, 588, 683, 1246, 609 |
| CIFAR-10 | 0.01 | 40 | 0, 3133, 0, 274, 373, 0, 296, 1709, 1114, 4636, 5287, 0, 1469, 1684, 0, 415, 3, 3950, 718, 46, 3355, 29, 7318, 1033, 0, 0, 918, 0, 1342, 0, 1, 0, 4979, 101, 1397, 1615, 5, 2798, 2, 0 |

achieve nearly identical accuracy, indicating that FedPTR is insensitive to the specific noise distribution. Local-random init (real local images without class matching) reaches 69.00, within 0.04 of class-aware init (69.04), which indicates that the gain over noise-based initialization comes mainly from using real local images rather than from matching classes. This robustness also supports the design of server-side FedPTR (FedPTR-S) that performs MTT and optimizes the auxiliary dataset with Gaussian initialization on the global server.

Table 9: Comparisons of different strategies of auxiliary dataset initialization on CIFAR-10 with Dirichlet $\alpha = 0.01$.

| Initialization strategy | CIFAR-10 |
|-------------------------|----------|
| FedPTR (zero init) | 65.12 |
| FedPTR (Gaussian init) | 67.50 |
| FedPTR (uniform init) | 68.51 |
| FedPTR (local-random init) | 69.00 |
| FedPTR (class-aware init, ours) | 69.04 |

## B.2 Reducing the Updates of the Auxiliary Dataset

For the update frequency, we have conducted ablation studies on client side as shown in Figure 2. Following the same setting, here we present our experiments on server side in Table 10, which shows that the server-side method FedPTR-S can still maintain a relatively stable performance with reduced updates of auxiliary datasets.

Table 10: Comparisons of FedPTR-S with different update frequencies of the auxiliary dataset on CIFAR-10 with Dirichlet $\alpha = 0.01$.

| Update Frequency | Test Acc |
|------------------|----------|
| per 1 round | 68.16 |
| per 4 rounds | 65.42 |
| per 10 rounds | 65.97 |
| per 40 rounds | 65.82 |
| one-shot | 65.34 |

## B.3 Size of the Auxiliary Dataset

We have conducted comparisons of the client's side method FedPTR with different sizes of the auxiliary dataset on CIFAR-10 with Dirichlet $\alpha = 0.01$ as shown in Table 4. Here we present the results on server

side method FedPTR-S with the same setting as shown in Table 11. We can observe FedPTR-S could obtain better performance with more auxiliary data, similar to FedPTR.

Table 11: Comparisons of FedPTR-S with different sizes of the auxiliary dataset on CIFAR-10 with Dirichlet $\alpha = 0.01$.

| Size of $\widetilde{\mathcal{D}}$ | Test Acc |
|---|---|
| 10 | 67.47 |
| 50 | 67.64 |
| 100 | 68.16 |
| 150 | 69.11 |
| 200 | 69.75 |

### B.4    Layer-Adaptive vs. Fixed Regularization

**Layer-adaptive $\lambda$ versus Fixed $\lambda$.** In Table 1, we have already shown the effectiveness of layer-adaptive $\lambda$, and here we give a further analysis. As shown in Figure 6, we present the complete convergence plots of FedPTR with layer-adaptive $\lambda$ and fixed $\lambda$ on CIFAR-10 and CIFAR-100 with Dirichlet $\alpha = 0.01$ and Dirichlet $\alpha = 0.04$. It can be seen that adopting layer-adaptive $\lambda$ slightly falls behind at the beginning of the training but obtains better performance in the late phase of the training, indicating that layer-adaptive $\lambda$ can help converge to a better stationary point.

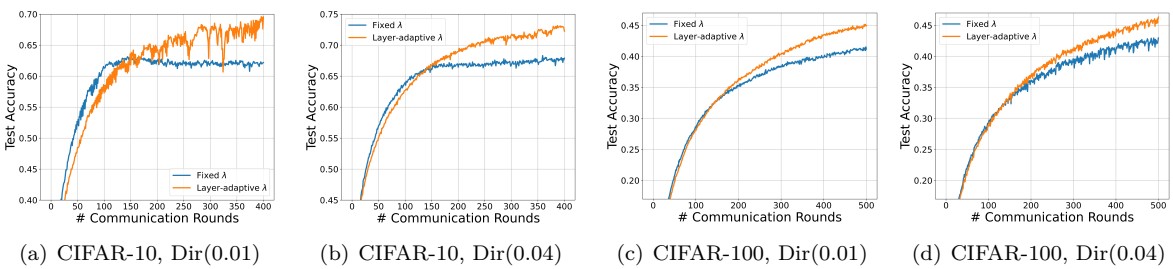

(a) CIFAR-10, Dir(0.01)    (b) CIFAR-10, Dir(0.04)    (c) CIFAR-100, Dir(0.01)    (d) CIFAR-100, Dir(0.04)

Figure 6: Test accuracy on CIFAR-10 and CIFAR-100 against the communication rounds, obtained by FedPTR with different regularizations.

### B.5    TinyImageNet Results

We additionally conduct experiments to test our method on larger TinyImageNet data to demonstrate that our method can indeed extend to larger-scale experiments. We summarize the experimental results with 10 clients and Dir $\alpha = 0.01$ in Table 12. The results suggest that our method still outperforms other baselines.

Table 12: Comparison of test accuracy for FedPTR (FedPTR-S) and other baselines on TinyImageNet.

| FedAvg | FedProx | Scaffold | FedDyn | FedDC | FedPTR | FedPTR-S |
|---|---|---|---|---|---|---|
| 27.68 | 27.11 | 21.55 | 27.49 | 23.81 | 28.6 | **29.6** |

### B.6    Ablation on the Trajectory Window Length $m$

The lookback window length $m$ controls how far back FedPTR looks into the global training trajectory when constructing the auxiliary dataset via MTT. To evaluate the sensitivity of FedPTR to this hyperparameter, we conduct an ablation across four (dataset $\times$ heterogeneity) combinations with $m \in \{3, 5, 10\}$, keeping all other settings identical to the main paper. As shown in Table 13, the variation across $m \in \{3, 5, 10\}$ is

bounded within ∼1.5 points across all four settings. Notably, in two of the four settings (CIFAR-10 with Dirichlet $\alpha = 0.01$ and CIFAR-100 with Dirichlet $\alpha = 0.04$), $m = 10$ outperforms the $m = 3$ default by 1.17 and 0.46 points respectively, while in the other two settings $m = 10$ stays within 1.7 points of $m = 3$. This indicates that FedPTR is robust to the choice of $m$, and that MTT continues to extract meaningful trajectory information even at a longer lookback gap, with the longer trajectory sometimes providing additional gain.

Table 13: Comparisons of FedPTR with different lookback window lengths $m$ on CIFAR-10 and CIFAR-100 with Dirichlet $\alpha = 0.01$ and $\alpha = 0.04$.

| Setting | $m = 3$ | $m = 5$ | $m = 10$ |
|---|---|---|---|
| CIFAR-10 Dir($\alpha = 0.01$) | 69.04 | 69.83 | 70.21 |
| CIFAR-10 Dir($\alpha = 0.04$) | 72.78 | 70.85 | 71.09 |
| CIFAR-100 Dir($\alpha = 0.01$) | 44.98 | 44.27 | 44.40 |
| CIFAR-100 Dir($\alpha = 0.04$) | 45.87 | 46.20 | 46.33 |

### B.7 Comparison with Linearly-Extrapolated Proximal-Center Baseline

To assess whether FedPTR's projected proximal center is meaningfully different from a naive linear extrapolation of past global models, we evaluate a FedProx + linear extrapolation baseline. The server-side anchor is computed as $\text{anchor}^t = w^t + \rho \cdot (w^t - w^{t-1})$ and pushed to all clients each round (falling back to $w^t$ in the first round), and the client local objective becomes $\arg\min_w \mathcal{L}(w; \mathcal{D}_i) + \frac{\lambda}{2}\|w - \text{anchor}^t\|^2$. We test two extrapolation strengths, $\rho = 0.5$ (half-step) and $\rho = 1.0$ (full-step). All other settings (CIFAR-10, Dirichlet $\alpha = 0.01$, 10 clients, 100% participation) are identical to the main paper. As shown in Table 14, linear extrapolation does provide a modest +3.24-point improvement at $\rho = 0.5$ but degrades at $\rho = 1.0$, since under extreme heterogeneity the aggregated trajectory direction $w^t - w^{t-1}$ is itself noisy and aggressive extrapolation amplifies that noise. FedPTR substantially outperforms the best linear-extrap baseline by +9.86 points, indicating that the gain comes not from anchor displacement per se but from the nonlinear, data-driven trajectory projection through MTT.

Table 14: Comparisons with FedProx + linear-extrapolation baseline on CIFAR-10 with Dirichlet $\alpha = 0.01$.

| Method | CIFAR-10 |
|---|---|
| FedProx (paper Table 2) | 55.94 |
| FedProx + linear extrap ($\rho = 0.5$) | 59.18 |
| FedProx + linear extrap ($\rho = 1.0$) | 57.73 |
| FedPTR (paper Table 2) | 69.04 |

## C Practical Considerations, Privacy, and Limitations

### C.1 Memory and Computation Overhead

We provide a quantitative accounting of FedPTR's per-client memory and computation cost on CIFAR-10 with Dirichlet $\alpha = 0.01$, measured on a single NVIDIA A6000.

**Memory.** FedPTR does not require clients to maintain all past global model states. The algorithm only needs the most recent $(m + 1)$ global model snapshots to be retained in a sliding buffer, which can be kept on the server side and pushed to clients only at the start of each MTT step. Consequently, each client only carries two additional persistent items beyond standard FL methods: the projected next-step model $\widetilde{w}_i^{t+1}$ (1× model size) and the auxiliary dataset $\widetilde{\mathcal{D}}_i$. For FedPTR-S, the auxiliary dataset is moved to the server, so the client-side overhead matches FedDyn's. The size of $\widetilde{\mathcal{D}}_i$ is a fixed hyperparameter independent of training horizon $T$ and model dimension; for CIFAR-10 with $|\widetilde{\mathcal{D}}_i| = 100$ images, it amounts to 1.17 MB. Table 15 summarizes byte-level measurements.

**Computation.** Per-client wall-clock per communication round on a single A6000 is 1.26 s for FedAvg, 1.29 s for FedProx, and 6.26 s for FedPTR with MTT executed every round (a 4.97× overhead). MTT updates,

Table 15: Per-client persistent memory on CIFAR-10 (ConvNet).

| Method | Per-client memory | Components |
|---|---|---|
| FedAvg | 1.22 MB | $w$ only |
| FedDyn | 2.44 MB | $w$ + local gradient state |
| FedPTR | 3.61 MB | $w$ + projected next-step model + $\widetilde{\mathcal{D}}_i$ (1.17 MB) |
| FedPTR-S | 2.44 MB | $w$ + projected next-step model (server-pushed) |

Table 16: Per-round per-client wall-clock and accuracy on CIFAR-10, Dir($\alpha = 0.01$), 400 rounds.

| Method | Per-client wall-clock | $\times$ FedAvg | Acc (last-5) |
|---|---|---|---|
| FedAvg | 1.26 s | 1.00$\times$ | 55.43 |
| FedProx | 1.29 s | 1.02$\times$ | 55.94 |
| FedPTR (MTT every round) | 6.26 s | 4.97$\times$ | 69.04 |
| FedPTR (MTT per 4 rounds) | 2.51 s | 1.99$\times$ | 66.87 |
| FedPTR (MTT per 10 rounds) | 1.79 s | 1.42$\times$ | 66.20 |
| FedPTR (MTT per 40 rounds) | 1.43 s | 1.13$\times$ | 65.74 |
| FedPTR (one-shot MTT) | 1.33 s | 1.06$\times$ | 64.67 |

however, need not be performed every round: as discussed in Section 6.3, reducing the MTT update frequency preserves accuracy well, and amortizing the MTT cost over multiple rounds drastically reduces the effective per-round overhead. Table 16 reports per-client wall-clock and final accuracy for FedPTR with several MTT update intervals. In particular, FedPTR with one-shot MTT runs at essentially FedAvg's wall-clock (1.06$\times$) while still outperforming FedDyn (62.29) by 2.4 points.

**Time to target accuracy.** We further report the per-client wall-clock required to first reach a given accuracy target (Table 17). Reduced-frequency FedPTR variants reach 50%/55% accuracy 2–3$\times$ faster than FedAvg or FedProx; FedPTR is the only method that reaches 60%+ accuracy under Dir($\alpha = 0.01$). For FedAvg and FedProx we use learning rate 0.005 (found to enable both baselines to surpass 50% accuracy by the end of training, at the cost of slower early convergence than the $lr = 0.01$ setting reported in Table 2).

Overall, these memory and computation comparisons clearly demonstrate that FedPTR introduces only modest overhead while delivering substantially higher accuracy, validating its practical viability.

## C.2 Limitation

The main limitation of FedPTR lies in its current scope of applicability rather than the federated framework itself. FedPTR relies on Matching Training Trajectories (MTT) (Cazenavette et al., 2022) to extract global information from the recent model training trajectory, and MTT requires the synthetic data to be optimized via gradient-based methods, which in turn requires the inputs to be differentiable. This restricts FedPTR to modalities with continuous-valued inputs (e.g., images), and prevents direct application to domains such as natural language, where inputs are sequences of discrete tokens. Some recent work tries to sidestep token-level non-differentiability by distilling auxiliary signals such as soft labels (Sucholutsky & Schonlau, 2021) or attention labels (Maekawa et al., 2023), but these efforts remain limited to small models and short-text classification. In particular, no MTT-style trajectory matching method has been reported for text in recent work (Maekawa et al., 2024). We view extending FedPTR to language and other modalities as an important future direction, which we expect to become tractable as trajectory-based dataset distillation matures beyond image inputs.

## C.3 Discussion of Assumption 5.3

Assumption 5.3 measures the gradient dissimilarity between the global loss function $\mathcal{L}(\mathbf{w}^t)$ and the auxiliary loss $\widetilde{\mathcal{L}}(\mathbf{w}; \widetilde{\mathcal{D}}_i)$ in expectation over the MTT randomness. This relaxation is analogous to the standard

Table 17: Per-client wall-clock to first reach a target accuracy on CIFAR-10, Dir($\alpha = 0.01$). N/A indicates the target was not reached within the 400-round training budget.

| Method | 50% | 55% | 60% | 65% |
|---|---|---|---|---|
| FedAvg | 205 s | 353 s | N/A | N/A |
| FedProx | 179 s | 311 s | N/A | N/A |
| FedPTR (every round) | 244 s | 344 s | 520 s | 983 s |
| FedPTR (per 4 rounds) | 103 s | 153 s | 233 s | 412 s |
| FedPTR (per 10 rounds) | 86 s | 125 s | 190 s | 394 s |
| FedPTR (per 40 rounds) | 67 s | 96 s | 173 s | 365 s |
| FedPTR (one-shot) | 63 s | 106 s | 176 s | 382 s |

bounded gradient variance treatment in prior FL literature. Crucially, the assumption no longer presumes that any individual auxiliary objective uniformly captures the global training dynamics; it only requires the MTT procedure to approximate the global gradient on average over its stochastic outputs, permitting larger point-wise deviation for individual realizations of $\mathcal{D}_i$.

We clarify the intended scope of Assumption 5.3 below:

- The quantity $\sigma_d^2$ bounds the gradient dissimilarity between the auxiliary loss $\widetilde{\mathcal{L}}(\mathbf{w}; \widetilde{\mathcal{D}}_i)$ and the global objective $\mathcal{L}(\mathbf{w}^t)$ evaluated at the same point. The number of the MTT inner steps, the trajectory time gap $m$ and other MTT algorithmic details influence the quality of $\widehat{\mathcal{D}}_i$, but they do not enter Assumption 5.3 as separate accumulating quantities. In other words, $\sigma_d^2$ captures whether the final $\widehat{\mathcal{D}}_i$ yields a useful auxiliary loss rather than how it was obtained step-by-step. Consequently, the residual $O(\eta^2 \sigma_d^2)$ in our convergence bound does not require characterization of the inner process of MTT.

- Assumption 5.3 and Theorem 5.4 are formulated entirely in terms of the auxiliary gradient, not in terms of trajectory-level properties such as path shape or uniqueness. Our framework does not claim to identify the canonical training trajectory. Rather, FedPTR uses the auxiliary dataset $\widetilde{\mathcal{D}}_i$ to produce a good update reference. The relevant question is therefore whether the selected auxiliary dataset yields a useful regularization signal for guiding the global model update. Formally, our framework only requires that MTT produces an auxiliary dataset whose induced gradient field is sufficiently aligned with the global gradient in expectation. This analytical choice is aligned with standard non-convex convergence analysis: results for SGD, FedAvg, FedAdam, and other FL methods aim to certify convergence to a stationary point (typically via bounds on $\min_t \mathbb{E}\|\nabla \mathcal{L}(\mathbf{w}^t)\|^2$), without characterizing which trajectory the optimizer takes. Our analysis follows this established convention and characterizes the key property that FedPTR's convergence depends on: when the auxiliary gradient provides a useful local correction signal.

- To empirically validate Assumption 5.3, we follow a similar experimental setup as in Figure 3. On the server side, we use the global update direction $\mathbf{d}_{\text{global}} = \mathbf{w}^t - \mathbf{w}^{t+1}$ as a proxy for the global gradient direction, and $\mathbf{d}_{\text{aux}} = \mathbf{w}^t - \frac{1}{N}\sum_i \widetilde{\mathbf{w}}_i^{t+1}$ as a proxy for the auxiliary gradient direction. We then measure $\|\mathbf{d}_{\text{global}} - \mathbf{d}_{\text{aux}}\|_2$ throughout training as an empirical proxy for $\sigma_d$. The trajectory of $\|\mathbf{d}_{\text{global}} - \mathbf{d}_{\text{aux}}\|_2$ across rounds provides an empirical estimate of $\mathbb{E}_{\mathcal{D}_i}[\|\nabla \mathcal{L}(\mathbf{w}) - \nabla \widetilde{\mathcal{L}}(\mathbf{w}; \widetilde{\mathcal{D}}_i)\|_2^2]$ across multiple realizations of the MTT randomness. Figure 7 presents the result on CIFAR-10 with Dirichlet(0.01). We observe that $\|\mathbf{d}_{\text{global}} - \mathbf{d}_{\text{aux}}\|_2$ ranges from approximately 2-3 in early training rounds to 6-8 in the last few training rounds. This confirms that $\sigma_d$ remains bounded in practice, consistent with Assumption 5.3. Moreover, the fact that this quantity remains bounded across hundreds of rounds provides direct empirical evidence that FedPTR faithfully tracks the global trajectory over long horizons.

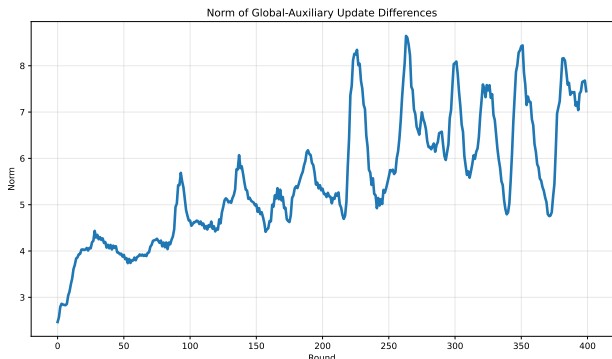

Figure 7: Empirical proxy for $\sigma_d$ throughout training on CIFAR-10 with Dirichlet(0.01). We plot $\|\mathbf{d}_{\text{global}} - \mathbf{d}_{\text{aux}}\|_2$, where $\mathbf{d}_{\text{global}} = \mathbf{w}^t - \mathbf{w}^{t+1}$ and $\mathbf{d}_{\text{aux}} = \mathbf{w}^t - \widetilde{\mathbf{w}}^{t+1}$. The deviation remains bounded throughout training.

## D  Proof of Theorem 5.4.

For notational convenience, we denote $\mathcal{L}(\mathbf{w}; \mathcal{D}_i)$ by $\mathcal{L}_i(\mathbf{w})$ and $L(\mathbf{w}; \widetilde{\mathcal{D}}_i)$ as $\widetilde{\mathcal{L}}_i(\mathbf{w})$ in the following.

**Lemma D.1.** For $h_i(\mathbf{w}) := \mathcal{L}_i(\mathbf{w}) + \frac{\lambda}{2}\|\mathbf{w} - \mathbf{w}^t + \eta\nabla\widetilde{\mathcal{L}}_i(\mathbf{w}^t)\|^2$, there is $\|\nabla h_i(\mathbf{w}) - \nabla h_i(\mathbf{v})\| \leq (L + \lambda)\|\mathbf{w} - \mathbf{v}\|, \forall \mathbf{w}, \mathbf{v} \in \mathbb{R}^d$.

*Proof of Lemma D.1.*

$$\|\nabla h_i(\mathbf{w}) - \nabla h_i(\mathbf{y})\|$$
$$= \|\nabla\mathcal{L}_i(\mathbf{w}) - \nabla\mathcal{L}_i(\mathbf{y}) + \lambda\mathbf{w} - \lambda\mathbf{y}\|$$
$$\leq \|\nabla\mathcal{L}_i(\mathbf{w}) - \nabla\mathcal{L}_i(\mathbf{y})\| + \lambda\|\mathbf{w} - \mathbf{y}\|$$
$$\leq (L + \lambda)\|\mathbf{w} - \mathbf{y}\|. \tag{D.1}$$

This concludes the proof. □

*Proof of Theorem 5.4.* Recall the notation and the objective function, we have the following for the steps,

$$\mathbf{w}_i^{t,q+1} = \mathbf{w}_i^{t,q} - \eta(\mathbf{g}_i^{t,q} + \lambda(\mathbf{w}_i^{t,q} - \widetilde{\mathbf{w}}_i^{t+1}))$$
$$= \mathbf{w}_i^{t,q} - \eta(\mathbf{g}_i^{t,q} + \lambda(\mathbf{w}_i^{t,q} - \mathbf{w}_t + \eta\nabla\widetilde{\mathcal{L}}_i(\mathbf{w}^t))), \tag{D.2}$$

where $q = 0, 1, ..., Q-1$ indicates the local step. We also have

$$h_i(\mathbf{w}) = f_i(\mathbf{w}) + \frac{\lambda}{2}\|\mathbf{w} - \mathbf{w}^t + \eta\nabla\widetilde{\mathcal{L}}_i(\mathbf{w}^t)\|^2,$$
$$\nabla h_i(\mathbf{w}) = \nabla\mathcal{L}_i(\mathbf{w}) + \lambda(\mathbf{w} - \mathbf{w}_t + \eta\nabla\widetilde{\mathcal{L}}_i(\mathbf{w}^t)). \tag{D.3}$$

Denote $\mathbf{e}_i^{t,q+1}$ as follows

$$\mathbf{e}_i^{t,q+1} = \nabla\mathcal{L}_i(\mathbf{w}_i^{t,q}) + \lambda(\mathbf{w}_i^{t,q} - \mathbf{w}_t + \eta\nabla\widetilde{\mathcal{L}}_i(\mathbf{w}^t)) = \nabla h_i(\mathbf{w}_i^{t,q}), \tag{D.4}$$

by the notation of $\gamma$-inexactness, there is

$$\|\mathbf{e}_i^{t,q+1}\|^2 \leq \gamma^2\|\nabla\mathcal{L}_i(\mathbf{w}^t - \eta\nabla\widetilde{\mathcal{L}}_i(\mathbf{w}^t))\|^2$$
$$= \gamma^2\|\nabla\mathcal{L}_i(\widetilde{\mathbf{w}}_i^{t+1})\|^2$$
$$= \gamma^2\|\nabla\mathcal{L}_i(\widetilde{\mathbf{w}}_i^{t+1}) - \nabla\mathcal{L}_i(\mathbf{w}^t) + \nabla\mathcal{L}_i(\mathbf{w}^t)\|^2$$
$$\leq 2\gamma^2\|\nabla\mathcal{L}_i(\mathbf{w}^t)\|^2 + 2\gamma^2 L^2\|\widetilde{\mathbf{w}}_i^{t+1} - \mathbf{w}^t\|^2$$
$$= 2\gamma^2\|\nabla\mathcal{L}_i(\mathbf{w}^t)\|^2 + 2\gamma^2 L^2\eta^2\|\nabla\widetilde{\mathcal{L}}_i(\mathbf{w}^t)\|^2, \tag{D.5}$$

where the first equation holds by the definition of distilled model weight $\widetilde{\mathbf{w}}_i^{t+1}$, the second inequality holds by the $L$-smoothness of $f_i$, and the last equation is from the definition of $\widetilde{\mathbf{w}}_i^{t+1}$. Then we have

$$\mathbb{E}_t[\mathbf{w}^{t+1} - \mathbf{w}^t] = \frac{1}{N} \sum_{i \in [N]} \mathbb{E}_t[\mathbf{w}_i^{t,Q} - \mathbf{w}^t]$$

$$= -\frac{1}{\lambda} \frac{1}{N} \sum_{i \in [N]} \mathbb{E}_t[\mathbf{g}_i^{t,Q} + \lambda\eta\nabla\widetilde{\mathcal{L}}_i(\mathbf{w}^t) - \mathbf{e}_i^{t,Q}]. \tag{D.6}$$

Define $\mu = \lambda - L_- > 0$ $[L_-]$ and $\mathbf{w}_{i,*}^{t+1} = \arg\min_{\mathbf{w}} h_i(\mathbf{w}; \mathbf{w}^t)$, then we have

$$\|\mathbf{w}_{i,*}^{t+1} - \mathbf{w}_i^{t,Q}\| \leq \frac{1}{\mu}\|\nabla h_i(\mathbf{w}_{i,*}^{t+1}) - \nabla h_i(\mathbf{w}_i^{t,Q})\|$$

$$= \frac{1}{\mu}\|\nabla h_i(\mathbf{w}_i^{t,Q})\|$$

$$\leq \frac{\gamma}{\mu}\|\nabla h_i(\widetilde{\mathbf{w}}_i^{t+1})\|$$

$$\leq \frac{\gamma}{\mu}\|\nabla h_i(\widetilde{\mathbf{w}}_i^{t+1}) - \nabla h_i(\mathbf{w}^t)\| + \frac{\gamma}{\mu}\|\nabla h_i(\mathbf{w}^t)\|$$

$$\leq \frac{\gamma}{\mu}(L+\lambda)\|\widetilde{\mathbf{w}}_i^{t+1} - \mathbf{w}^t\| + \frac{\gamma}{\mu}\|\nabla\mathcal{L}_i(\mathbf{w}^t) + \lambda\eta\nabla\widetilde{\mathcal{L}}_i(\mathbf{w}^t)\|$$

$$= \frac{\gamma\eta}{\mu}(L+\lambda)\|\nabla\widetilde{\mathcal{L}}_i(\mathbf{w}^t)\| + \frac{\gamma}{\mu}\|\nabla\mathcal{L}_i(\mathbf{w}^t) + \lambda\eta\nabla\widetilde{\mathcal{L}}_i(\mathbf{w}^t)\|, \tag{D.7}$$

where the first inequality holds by the $\mu$-strong convexity of $h_i$, the first equation holds by $\nabla h_i(\mathbf{w}_{i,*}^{t+1}) = 0$ since $\mathbf{w}_{i,*}^{t+1} = \arg\min_{\mathbf{w}} h_i(\mathbf{w}; \mathbf{w}^t)$, and the second inequality holds by $\gamma$-inexactness. The third inequality follows the triangle inequality, the forth one follows Lemma D.1. Similarly by the $\mu$-strongly convexity, we have

$$\|\mathbf{w}_{i,*}^{t+1} - \mathbf{w}^t\| \leq \frac{1}{\mu}\|\nabla h_i(\mathbf{w}^t)\| = \frac{1}{\mu}\|\nabla\mathcal{L}_i(\mathbf{w}^t) + \lambda\eta\widetilde{\mathcal{L}}_i(\mathbf{w}^t)\|. \tag{D.8}$$

By the triangle inequality, we have

$$\|\mathbf{w}_i^{t,Q} - \mathbf{w}^t\| \leq \|\mathbf{w}_{i,*}^{t+1} - \mathbf{w}^t\| + \|\mathbf{w}_{i,*}^{t+1} - \mathbf{w}_i^{t,Q}\|$$

$$\leq \frac{1+\gamma}{\mu}\|\nabla\mathcal{L}_i(\mathbf{w}^t) + \lambda\eta\nabla\widetilde{\mathcal{L}}_i(\mathbf{w}^t)\| + \frac{\gamma\eta(L+\lambda)}{\mu}\|\nabla\widetilde{\mathcal{L}}_i(\mathbf{w}^t)\| \tag{D.9}$$

and

$$\|\mathbf{w}_i^{t,Q} - \mathbf{w}^t\|^2 \leq 2\|\mathbf{w}_{i,*}^{t+1} - \mathbf{w}^t\|^2 + 2\|\mathbf{w}_{i,*}^{t+1} - \mathbf{w}_i^{t,Q}\|^2$$

$$\leq \frac{2+4\gamma^2}{\mu^2}\|\nabla\mathcal{L}_i(\mathbf{w}^t) + \lambda\eta\nabla\widetilde{\mathcal{L}}_i(\mathbf{w}^t)\|^2 + \frac{4\gamma^2\eta^2(L+\lambda)^2}{\mu^2}\|\nabla\widetilde{\mathcal{L}}_i(\mathbf{w}^t)\|^2. \tag{D.10}$$

Since the objective function $f$ is $L$ smooth, taking conditional expectation at time $t$, (abbreviate $\mathbb{E}_t$ as $\mathbb{E}$) we have

$$\mathbb{E}_t[\mathcal{L}(\mathbf{w}^{t+1})] \leq \mathcal{L}(\mathbf{w}_t) + \mathbb{E}_t\langle\nabla\mathcal{L}(\mathbf{w}_t), \mathbf{w}^{t+1} - \mathbf{w}^t\rangle + \frac{L}{2}\mathbb{E}_t[\|\mathbf{w}^{t+1} - \mathbf{w}^t\|^2]$$

$$\leq \mathcal{L}(\mathbf{w}_t) + \underbrace{\mathbb{E}\left\langle\nabla\mathcal{L}(\mathbf{w}_t), -\frac{1}{\lambda}\frac{1}{N}\sum_{i\in[N]}[\mathbf{g}_i^{t,Q} + \lambda\eta\nabla\widetilde{\mathcal{L}}_i(\mathbf{w}^t) - \mathbf{e}_i^{t,Q}]\right\rangle}_{I_1} + \underbrace{\frac{L}{2}\mathbb{E}[\|\mathbf{w}^{t+1} - \mathbf{w}^t\|^2]}_{I_2}. \tag{D.11}$$

For term $I_2$, we have

$$
\begin{aligned}
I_2 &= \frac{L}{2}\mathbb{E}_t[\|\mathbf{w}^{t+1} - \mathbf{w}^t\|^2] \\
&= \frac{L}{2}\mathbb{E}\left[\left\|\frac{1}{N}\sum_{i\in[N]}\mathbf{w}_i^{t,Q} - \mathbf{w}^t\right\|^2\right] \\
&\leq \frac{L}{2}\frac{1}{N}\sum_{i\in[N]}\mathbb{E}[\|\mathbf{w}_i^{t,Q} - \mathbf{w}^t\|^2] \\
&\leq \frac{1+2\gamma^2}{\mu^2}\frac{L}{N}\sum_{i\in[N]}\mathbb{E}[\|\nabla\mathcal{L}_i(\mathbf{w}^t) + \lambda\eta\nabla\widetilde{\mathcal{L}}_i(\mathbf{w}^t)\|^2] + \frac{4\gamma^2\eta^2L^2}{\mu^2}\frac{L}{N}\sum_{i\in[N]}\mathbb{E}[\|\nabla\widetilde{\mathcal{L}}_i(\mathbf{w}^t)\|^2] \\
&\leq \frac{2+4\gamma^2}{\mu^2}\frac{L}{N}\sum_{i\in[N]}\mathbb{E}[\|\nabla\mathcal{L}_i(\mathbf{w}^t)\|^2] + \left(\frac{4\gamma^2\eta^2(L+\lambda)^2}{\mu^2} + \frac{(2+4\gamma^2)\lambda^2\eta^2}{\mu^2}\right)\frac{L}{N}\sum_{i\in[N]}\mathbb{E}[\|\nabla\widetilde{\mathcal{L}}_i(\mathbf{w}^t)\|^2] \\
&\leq \frac{LB^2(2+4\gamma^2)}{\mu^2}\mathbb{E}[\|\nabla\mathcal{L}(\mathbf{w}^t)\|^2] \\
&\quad + \left(\frac{4\gamma^2\eta^2(L+\lambda)^2}{\mu^2} + \frac{(2+4\gamma^2)\lambda^2\eta^2}{\mu^2}\right)\frac{L}{N}\sum_{i\in[N]}\mathbb{E}[\|\nabla\widetilde{\mathcal{L}}_i(\mathbf{w}^t) - \nabla\mathcal{L}(\mathbf{w}^t) + \nabla\mathcal{L}(\mathbf{w}^t)\|^2], \quad\quad (\text{D.12})
\end{aligned}
$$

where the first and third inequalities hold by Cauchy inequality, the forth one holds by Assumption 5.2, and the last one holds by Assumption 5.3. Then we have

$$
\begin{aligned}
&\left(\frac{4\gamma^2\eta^2(L+\lambda)^2}{\mu^2} + \frac{(2+4\gamma^2)\lambda^2\eta^2}{\mu^2}\right)\frac{L}{N}\sum_{i\in[N]}\mathbb{E}[\|\nabla\widetilde{\mathcal{L}}_i(\mathbf{w}^t) - \nabla\mathcal{L}(\mathbf{w}^t) + \nabla\mathcal{L}(\mathbf{w}^t)\|^2] \\
&\leq 2\left(\frac{4\gamma^2\eta^2(L+\lambda)^2}{\mu^2} + \frac{(2+4\gamma^2)\lambda^2\eta^2}{\mu^2}\right)\frac{L}{N}\sum_{i\in[N]}\mathbb{E}\{\mathbb{E}_{\mathcal{D}_i}[\|\nabla\widetilde{\mathcal{L}}_i(\mathbf{w}^t) - \nabla\mathcal{L}(\mathbf{w}^t)\|^2]\} \\
&\quad + 2\left(\frac{4\gamma^2\eta^2(L+\lambda)^2}{\mu^2} + \frac{(2+4\gamma^2)\lambda^2\eta^2}{\mu^2}\right)\frac{L}{N}\sum_{i\in[N]}\mathbb{E}[\|\nabla\mathcal{L}(\mathbf{w}^t)\|^2] \\
&\leq \left(\frac{8\gamma^2\eta^2(L+\lambda)^2}{\mu^2} + \frac{(4+8\gamma^2)\lambda^2\eta^2}{\mu^2}\right)L\sigma_d^2 + \left(\frac{8\gamma^2\eta^2(L+\lambda)^2}{\mu^2} + \frac{(4+8\gamma^2)\lambda^2\eta^2}{\mu^2}\right)L\mathbb{E}[\|\nabla\mathcal{L}(\mathbf{w}^t)\|^2]. \quad (\text{D.13})
\end{aligned}
$$

Thus

$$
\begin{aligned}
I_2 &\leq \left(\frac{B^2(2+4\gamma^2)}{\mu^2} + \frac{8\gamma^2\eta^2(L+\lambda)^2}{\mu^2} + \frac{(4+8\gamma^2)\lambda^2\eta^2}{\mu^2}\right)L\mathbb{E}[\|\nabla\mathcal{L}(\mathbf{w}^t)\|^2] \\
&\quad + \left(\frac{8\gamma^2\eta^2(L+\lambda)^2}{\mu^2} + \frac{(4+8\gamma^2)\lambda^2\eta^2}{\mu^2}\right)L\sigma_d^2. \quad\quad (\text{D.14})
\end{aligned}
$$

Similarly, it also implies

$$
\begin{aligned}
\mathbb{E}[\|\mathbf{w}^{t+1} - \mathbf{w}^t\|^2] &\leq \frac{1}{N}\sum_{i\in[N]}\mathbb{E}[\|\mathbf{w}_i^{t+1} - \mathbf{w}^t\|^2] \\
&\leq 2\left(\frac{B^2(2+4\gamma^2)}{\mu^2} + \frac{8\gamma^2\eta^2(L+\lambda)^2}{\mu^2} + \frac{(4+8\gamma^2)\lambda^2\eta^2}{\mu^2}\right)\mathbb{E}[\|\nabla\mathcal{L}(\mathbf{w}^t)\|^2] \\
&\quad + 2\left(\frac{8\gamma^2\eta^2(L+\lambda)^2}{\mu^2} + \frac{(4+8\gamma^2)\lambda^2\eta^2}{\mu^2}\right)\sigma_d^2. \quad\quad (\text{D.15})
\end{aligned}
$$

For term $I_1$, we have

$$
\begin{aligned}
I_1 &= \mathbb{E}\left\langle \nabla\mathcal{L}(\mathbf{w}_t), -\frac{1}{\lambda}\frac{1}{N}\sum_{i\in[N]}[\mathbf{g}_i^{t,Q} + \lambda\eta\nabla\widetilde{\mathcal{L}}_i(\mathbf{w}^t) - \mathbf{e}_i^{t,Q}] + \frac{1}{\lambda}\nabla\mathcal{L}(\mathbf{w}^t) - \frac{1}{\lambda}\nabla\mathcal{L}(\mathbf{w}^t)\right\rangle \\
&= -\frac{1}{\lambda}\mathbb{E}[\|\nabla\mathcal{L}(\mathbf{w}_t)\|^2] + \mathbb{E}\left\langle \nabla\mathcal{L}(\mathbf{w}_t), -\frac{1}{\lambda}\frac{1}{N}\sum_{i\in[N]}[\nabla\mathcal{L}_i(\mathbf{w}_i^{t,Q}) + \lambda\eta\nabla\widetilde{\mathcal{L}}_i(\mathbf{w}^t) - \mathbf{e}_i^{t,Q}] + \frac{1}{\lambda}\nabla\mathcal{L}(\mathbf{w}^t)\right\rangle \\
&= -\frac{1}{\lambda}\mathbb{E}[\|\nabla\mathcal{L}(\mathbf{w}_t)\|^2] + \mathbb{E}\left\langle \nabla\mathcal{L}(\mathbf{w}_t), -\frac{1}{\lambda}\frac{1}{N}\sum_{i\in[N]}[\nabla\mathcal{L}_i(\mathbf{w}_i^{t,Q}) + \lambda\eta\nabla\widetilde{\mathcal{L}}_i(\mathbf{w}^t) - \mathbf{e}_i^{t,Q}] + \frac{1}{\lambda}\frac{1}{N}\sum_{i\in[N]}\nabla\mathcal{L}_i(\mathbf{w}^t)\right\rangle \\
&= -\frac{1}{\lambda}\mathbb{E}[\|\nabla\mathcal{L}(\mathbf{w}_t)\|^2] + \frac{1}{2\lambda}\mathbb{E}[\|\nabla\mathcal{L}(\mathbf{w}_t)\|^2] \\
&\quad + \underbrace{\frac{1}{2\lambda}\mathbb{E}\left[\left\|-\frac{1}{N}\sum_{i\in[N]}[\nabla\mathcal{L}_i(\mathbf{w}_i^{t,Q}) + \lambda\eta\nabla\widetilde{\mathcal{L}}_i(\mathbf{w}^t) - \mathbf{e}_i^{t,Q}] + \nabla\mathcal{L}(\mathbf{w}^t)\right\|^2\right]}_{I_3} \\
&\quad - \frac{1}{2\lambda}\mathbb{E}\left[\left\|-\frac{1}{N}\sum_{i\in[N]}[\nabla\mathcal{L}_i(\mathbf{w}_i^{t,Q}) + \lambda\eta\nabla\widetilde{\mathcal{L}}_i(\mathbf{w}^t) - \mathbf{e}_i^{t,Q}]\right\|^2\right],
\end{aligned}
\tag{D.16}
$$

where the second equation holds due to the unbiased-ness of stochastic gradient $\mathbf{g}_i^{t,Q}$, the third equation holds by the definition of global objective function, the forth one holds by the fact of $\langle \mathbf{a}, \boldsymbol{b}\rangle = \frac{1}{2}[\|\mathbf{a}\|^2 + \|\boldsymbol{b}\|^2 - \|\mathbf{a} - \boldsymbol{b}\|^2]$. For $I_3$ term, we have

$$
\begin{aligned}
I_3 &= \frac{1}{2\lambda}\mathbb{E}\left[\left\|-\frac{1}{N}\sum_{i\in[N]}\nabla\mathcal{L}_i(\mathbf{w}_i^{t,Q}) - \frac{1}{N}\sum_{i\in[N]}\lambda\eta\nabla\widetilde{\mathcal{L}}_i(\mathbf{w}^t) + \frac{1}{N}\sum_{i\in[N]}\mathbf{e}_i^{t,Q} + \frac{1}{N}\sum_{i\in[N]}\nabla\mathcal{L}_i(\mathbf{w}^t)\right\|^2\right] \\
&\leq \frac{1}{\lambda}\mathbb{E}\left[\left\|\frac{1}{N}\sum_{i\in[N]}\nabla\mathcal{L}_i(\mathbf{w}^t) - \frac{1}{N}\sum_{i\in[N]}\nabla\mathcal{L}_i(\mathbf{w}_i^{t,Q})\right\|^2\right] + \frac{1}{\lambda}\mathbb{E}\left[\left\|-\frac{1}{N}\sum_{i\in[N]}\lambda\eta\nabla\widetilde{\mathcal{L}}_i(\mathbf{w}^t) + \frac{1}{N}\sum_{i\in[N]}\mathbf{e}_i^{t,Q}\right\|^2\right] \\
&\leq \frac{L^2}{\lambda}\frac{1}{N}\sum_{i\in[N]}\mathbb{E}[\|\mathbf{w}_i^{t,Q} - \mathbf{w}^t\|^2] + \frac{2}{\lambda}\mathbb{E}\left[\left\|\frac{\lambda\eta}{N}\sum_{i\in[N]}\nabla\widetilde{\mathcal{L}}_i(\mathbf{w}^t)\right\|^2\right] + \frac{2}{\lambda}\frac{1}{N}\sum_{i\in[N]}\mathbb{E}[\|\mathbf{e}_i^{t,Q}\|^2] \\
&\leq \frac{L^2}{\lambda}\Bigg[\left(\frac{B^2(4+8\gamma^2)}{\mu^2} + \frac{16\gamma^2\eta^2(L+\lambda)^2}{\mu^2} + \frac{(8+16\gamma^2)\lambda^2\eta^2}{\mu^2}\right)\mathbb{E}[\|\nabla\mathcal{L}(\mathbf{w}^t)\|^2] \\
&\quad + \left(\frac{16\gamma^2\eta^2(L+\lambda)^2}{\mu^2} + \frac{(8+16\gamma^2)\lambda^2\eta^2}{\mu^2}\right)\sigma_d^2\Bigg] \\
&\quad + \frac{2\lambda\eta^2}{N}\sum_{i\in[N]}\mathbb{E}[\|\nabla\widetilde{\mathcal{L}}_i(\mathbf{w}^t)\|^2] + \frac{2}{\lambda}\frac{1}{N}\sum_{i\in[N]}[2\gamma^2\mathbb{E}[\|\nabla\mathcal{L}_i(\mathbf{w}^t)\|^2] + 2\gamma^2L^2\eta^2\mathbb{E}[\|\nabla\widetilde{\mathcal{L}}_i(\mathbf{w}^t)\|^2]] \\
&\leq \frac{L^2}{\lambda}\Bigg[\left(\frac{B^2(4+8\gamma^2)}{\mu^2} + \frac{16\gamma^2\eta^2(L+\lambda)^2}{\mu^2} + \frac{(8+16\gamma^2)\lambda^2\eta^2}{\mu^2}\right)\mathbb{E}[\|\nabla\mathcal{L}(\mathbf{w}^t)\|^2] \\
&\quad + \left(\frac{16\gamma^2\eta^2(L+\lambda)^2}{\mu^2} + \frac{(8+16\gamma^2)\lambda^2\eta^2}{\mu^2}\right)\sigma_d^2\Bigg] \\
&\quad + \left(\frac{2\lambda\eta^2}{N} + \frac{4\gamma^2L^2\eta^2}{\lambda N}\right)\sum_{i\in[N]}\mathbb{E}[\|\nabla\widetilde{\mathcal{L}}_i(\mathbf{w}^t)\|^2] + \frac{4\gamma^2}{\lambda}\frac{1}{N}\sum_{i\in[N]}\mathbb{E}[\|\nabla\mathcal{L}_i(\mathbf{w}^t)\|^2],
\end{aligned}
\tag{D.17}
$$

where the first and last inequalities hold by Cauchy inequality, and the second one holds by $L$-smoothness and Cauchy inequality as well. The third one is due to the bound for $\|\mathbf{w}_i^{t,Q} - \mathbf{w}^t\|^2$ and the $\gamma$-inexactness of

$\mathbf{e}_i^{t,Q}$. Then by the bounded gradient dissimilarity and bounded distilled gradient, organizing pieces, we have

$$
\begin{aligned}
I_3 &\leq \frac{L^2}{\lambda}\left[\left(\frac{B^2(4+8\gamma^2)}{\mu^2} + \frac{16\gamma^2\eta^2(L+\lambda)^2}{\mu^2} + \frac{(8+16\gamma^2)\lambda^2\eta^2}{\mu^2}\right)\mathbb{E}[\|\nabla\mathcal{L}(\mathbf{w}^t)\|^2]\right. \\
&\quad \left. + \left(\frac{16\gamma^2\eta^2(L+\lambda)^2}{\mu^2} + \frac{(8+16\gamma^2)\lambda^2\eta^2}{\mu^2}\right)\sigma_d^2\right] \\
&\quad + \left(\frac{2\lambda\eta^2}{N} + \frac{4\gamma^2L^2\eta^2}{\lambda N}\right)\sum_{i\in[N]}\mathbb{E}[\|\nabla\widetilde{\mathcal{L}}_i(\mathbf{w}^t)\|^2] + \frac{4\gamma^2}{\lambda}\frac{1}{N}\sum_{i\in[N]}\mathbb{E}[\|\nabla\mathcal{L}_i(\mathbf{w}^t)\|^2] \\
&\leq \frac{L^2}{\lambda}\left[\left(\frac{B^2(4+8\gamma^2)}{\mu^2} + \frac{16\gamma^2\eta^2(L+\lambda)^2}{\mu^2} + \frac{(8+16\gamma^2)\lambda^2\eta^2}{\mu^2}\right)\mathbb{E}[\|\nabla\mathcal{L}(\mathbf{w}^t)\|^2]\right. \\
&\quad \left. + \left(\frac{16\gamma^2\eta^2(L+\lambda)^2}{\mu^2} + \frac{(8+16\gamma^2)\lambda^2\eta^2}{\mu^2}\right)\sigma_d^2\right] + \left(4\lambda\eta^2 + \frac{8\gamma^2L^2\eta^2}{\lambda}\right)\sigma_d^2 \\
&\quad + \left(\frac{4\lambda\eta^2}{N} + \frac{8\gamma^2L^2\eta^2}{\lambda N}\right)\sum_{i\in[N]}\mathbb{E}[\|\nabla\mathcal{L}(\mathbf{w}^t)\|^2] + \frac{4\gamma^2}{\lambda N}\sum_{i\in[N]}\mathbb{E}[\|\nabla\mathcal{L}_i(\mathbf{w}^t)\|^2] \\
&\leq \left[\frac{L^2}{\lambda}\left(\frac{B^2(4+8\gamma^2)}{\mu^2} + \frac{16\gamma^2\eta^2(L+\lambda)^2}{\mu^2} + \frac{(8+16\gamma^2)\lambda^2\eta^2}{\mu^2}\right) + \left(\frac{4\lambda\eta^2}{N} + \frac{8\gamma^2L^2\eta^2}{\lambda N}\right) + \frac{4\gamma^2}{\lambda}B^2\right]\mathbb{E}[\|\nabla\mathcal{L}(\mathbf{w}^t)\|^2] \\
&\quad + \left[\frac{L^2}{\lambda}\left(\frac{16\gamma^2\eta^2(L+\lambda)^2}{\mu^2} + \frac{(8+16\gamma^2)\lambda^2\eta^2}{\mu^2}\right) + \left(4\lambda\eta^2 + \frac{8\gamma^2L^2\eta^2}{\lambda}\right)\right]\sigma_d^2. \quad\text{(D.18)}
\end{aligned}
$$

Therefore, by merging pieces together, we have

$$
\begin{aligned}
&\mathbb{E}[\mathcal{L}(\mathbf{w}_{t+1})] - \mathbb{E}[\mathcal{L}(\mathbf{w}_t)] \\
&\leq \left[-\frac{1}{2\lambda} + \frac{L^2}{\lambda}\left(\frac{B^2(4+8\gamma^2)}{\mu^2} + \frac{16\gamma^2\eta^2(L+\lambda)^2}{\mu^2} + \frac{(8+16\gamma^2)\lambda^2\eta^2}{\mu^2}\right) + \frac{4\lambda\eta^2}{N} + \frac{8\gamma^2L^2\eta^2}{\lambda N} + \frac{4\gamma^2B^2}{\lambda}\right. \\
&\quad \left. + \frac{LB^2(2+4\gamma^2)}{\mu^2} + \frac{8L\gamma^2\eta^2(L+\lambda)^2}{\mu^2} + \frac{L(4+8\gamma^2)\lambda^2\eta^2}{\mu^2}\right]\mathbb{E}\|\nabla\mathcal{L}(\mathbf{w}^t)\|^2 \\
&\quad + \left[\frac{L^2}{\lambda}\left(\frac{16\gamma^2\eta^2(L+\lambda)^2}{\mu^2} + \frac{(8+16\gamma^2)\lambda^2\eta^2}{\mu^2}\right) + 4\lambda\eta^2 + \frac{8\gamma^2L^2\eta^2}{\lambda} + L\left(\frac{8\gamma^2\eta^2(L+\lambda)^2}{\mu^2} + \frac{(4+8\gamma^2)\lambda^2\eta^2}{\mu^2}\right)\right]\sigma_d^2, \\
&\hspace{14cm}\text{(D.19)}
\end{aligned}
$$

summing up from $t=0$ to $T-1$, we have

$$
\begin{aligned}
&\mathbb{E}[\mathcal{L}(\mathbf{w}_T)] - \mathbb{E}[\mathcal{L}(\mathbf{w}_0)] \\
&\leq \left[-\frac{1}{2\lambda} + \frac{L^2}{\lambda}\left(\frac{B^2(4+8\gamma^2)}{\mu^2} + \frac{16\gamma^2\eta^2(L+\lambda)^2}{\mu^2} + \frac{(8+16\gamma^2)\lambda^2\eta^2}{\mu^2}\right) + \frac{4\lambda\eta^2}{N} + \frac{8\gamma^2L^2\eta^2}{\lambda N} + \frac{4\gamma^2B^2}{\lambda}\right. \\
&\quad \left. + \frac{LB^2(2+4\gamma^2)}{\mu^2} + \frac{8L\gamma^2\eta^2(L+\lambda)^2}{\mu^2} + \frac{L(4+8\gamma^2)\lambda^2\eta^2}{\mu^2}\right]\sum_{t=0}^{T-1}\mathbb{E}[\|\nabla\mathcal{L}(\mathbf{w}^t)\|^2] \\
&\quad + T\left[\frac{L^2}{\lambda}\left(\frac{16\gamma^2\eta^2(L+\lambda)^2}{\mu^2} + \frac{(8+16\gamma^2)\lambda^2\eta^2}{\mu^2}\right) + 4\lambda\eta^2 + \frac{8\gamma^2L^2\eta^2}{\lambda} + L\left(\frac{8\gamma^2\eta^2(L+\lambda)^2}{\mu^2} + \frac{(4+8\gamma^2)\lambda^2\eta^2}{\mu^2}\right)\right]\sigma_d^2, \\
&\hspace{14cm}\text{(D.20)}
\end{aligned}
$$

then

$$\left[\frac{1}{2\lambda} - \frac{L^2}{\lambda}\left(\frac{B^2(4+8\gamma^2)}{\mu^2} + \frac{16\gamma^2\eta^2(L+\lambda)^2}{\mu^2} + \frac{(8+16\gamma^2)\lambda^2\eta^2}{\mu^2}\right) - \frac{4\lambda\eta^2}{N} - \frac{8\gamma^2L^2\eta^2}{\lambda N} - \frac{4\gamma^2B^2}{\lambda}\right.$$

$$\left.- \frac{LB^2(2+4\gamma^2)}{\mu^2} - \frac{8L\gamma^2\eta^2(L+\lambda)^2}{\mu^2} - \frac{L(4+8\gamma^2)\lambda^2\eta^2}{\mu^2}\right]\sum_{t=0}^{T-1}\mathbb{E}[\|\nabla\mathcal{L}(\mathbf{w}^t)\|^2]$$

$$\leq \mathbb{E}[\mathcal{L}(\mathbf{w}_0)] - \mathbb{E}[\mathcal{L}(\mathbf{w}_T)] + T\left[\frac{L^2}{\lambda}\left(\frac{16\gamma^2\eta^2(L+\lambda)^2}{\mu^2} + \frac{(8+16\gamma^2)\lambda^2\eta^2}{\mu^2}\right)\right.$$

$$\left.+ 4\lambda\eta^2 + \frac{8\gamma^2L^2\eta^2}{\lambda} + L\left(\frac{8\gamma^2\eta^2(L+\lambda)^2}{\mu^2} + \frac{(4+8\gamma^2)\lambda^2\eta^2}{\mu^2}\right)\right]\sigma_d^2, \tag{D.21}$$

hence we have

$$\frac{1}{T}\sum_{t=0}^{T-1}\mathbb{E}[\|\nabla\mathcal{L}(\mathbf{w}^t)\|^2]$$

$$\leq \left[\frac{1}{2\lambda} - \frac{L^2}{\lambda}\left(\frac{B^2(4+8\gamma^2)}{\mu^2} + \frac{16\gamma^2\eta^2(L+\lambda)^2}{\mu^2} + \frac{(8+16\gamma^2)\lambda^2\eta^2}{\mu^2}\right) - \frac{4\lambda\eta^2}{N} - \frac{8\gamma^2L^2\eta^2}{\lambda N} - \frac{4\gamma^2B^2}{\lambda}\right.$$

$$\left.- \frac{LB^2(2+4\gamma^2)}{\mu^2} - \frac{8L\gamma^2\eta^2(L+\lambda)^2}{\mu^2} - \frac{L(4+8\gamma^2)\lambda^2\eta^2}{\mu^2}\right]^{-1}\frac{\mathbb{E}[\mathcal{L}(\mathbf{w}_0)] - \mathbb{E}[\mathcal{L}(\mathbf{w}_T)]}{T}$$

$$+ \left[\frac{L^2}{\lambda}\left(\frac{16\gamma^2\eta^2(L+\lambda)^2}{\mu^2} + \frac{(8+16\gamma^2)\lambda^2\eta^2}{\mu^2}\right) + 4\lambda\eta^2 + \frac{8\gamma^2L^2\eta^2}{\lambda} + L\left(\frac{8\gamma^2\eta^2(L+\lambda)^2}{\mu^2} + \frac{(4+8\gamma^2)\lambda^2\eta^2}{\mu^2}\right)\right]\sigma_d^2. \tag{D.22}$$

If we assume that $\eta \leq \frac{1}{2\lambda}$ and $\gamma \to 0$, then we have

$$\left[\frac{1}{2\lambda} - \frac{L^2}{\lambda}\left(\frac{B^2(4+8\gamma^2)}{\mu^2} + \frac{16\gamma^2\eta^2(L+\lambda)^2}{\mu^2} + \frac{(8+16\gamma^2)\lambda^2\eta^2}{\mu^2}\right) - \frac{4\lambda\eta^2}{N} - \frac{8\gamma^2L^2\eta^2}{\lambda N} - \frac{4\gamma^2B^2}{\lambda}\right.$$

$$\left.- \frac{LB^2(2+4\gamma^2)}{\mu^2} - \frac{8L\gamma^2\eta^2(L+\lambda)^2}{\mu^2} - \frac{L(4+8\gamma^2)\lambda^2\eta^2}{\mu^2}\right]$$

$$= \frac{1}{2\lambda} - \frac{4L^2B^2}{\lambda\mu^2} - \frac{8L^2\lambda\eta^2}{\mu^2} - \frac{4\lambda\eta^2}{N} - \frac{2LB^2}{\mu^2} - \frac{4L\lambda^2\eta^2}{\mu^2} - \mathcal{O}(\gamma^2)$$

$$\geq \frac{1}{2\lambda} - \frac{4L^2B^2}{\lambda\mu^2} - \frac{2L^2}{\lambda\mu^2} - \frac{1}{\lambda N} - \frac{2LB^2}{\mu^2} - \frac{4L}{\mu^2} - \mathcal{O}(\gamma^2) > 0, \tag{D.23}$$

and

$$\frac{8L^2B^2\gamma^2}{\lambda\mu^2} + \frac{16\gamma^2L^2\eta^2(L+\lambda)^2}{\lambda\mu^2} + \frac{16\gamma^2L^2\eta^2\lambda^2}{\lambda\mu^2} + \frac{8\gamma^2L^2\eta^2}{\lambda N} + \frac{4\gamma^2B^2}{\lambda} + \frac{4\gamma^2LB^2}{\mu^2} + \frac{8L\gamma^2\eta^2(L+\lambda)^2}{\mu^2} + \frac{8L\gamma^2\lambda^2\eta^2}{\mu^2}$$

$$\leq \frac{8L^2B^2\gamma^2}{\lambda\mu^2} + \frac{4\gamma^2L^2(L+\lambda)^2}{\lambda^3\mu^2} + \frac{4\gamma^2L^2}{\lambda\mu^2} + \frac{2\gamma^2L^2}{\lambda^3N} + \frac{4\gamma^2B^2}{\lambda} + \frac{4\gamma^2LB^2}{\mu^2} + \frac{2L\gamma^2(L+\lambda)^2}{\lambda^2\mu^2} + \frac{2L\gamma^2}{\mu^2} = \mathcal{O}(\gamma^2), \tag{D.24}$$

hence we need the condition of

$$\mathcal{O}(\gamma^2) \leq \frac{1}{6\lambda} \text{ and } \frac{1}{6\lambda} - \frac{4L^2B^2}{\lambda\mu^2} - \frac{2L^2}{\lambda\mu^2} - \frac{1}{\lambda N} - \frac{2LB^2}{\mu^2} - \frac{4L}{\mu^2} \geq 0 \tag{D.25}$$

$$\gamma \leq 1/\sqrt{C\lambda}, \tag{D.26}$$

where $C$ is a numerical constant depending on $L, B, \gamma, \lambda, \eta$, etc. Reorganizing terms, we have

$$\min_{t\in[T]}\mathbb{E}[\|\nabla\mathcal{L}(\mathbf{w}_t)\|_2^2] \leq \mathcal{O}\left(\frac{\mathbb{E}[\mathcal{L}(\mathbf{w}_0)] - \mathbb{E}[\mathcal{L}(\mathbf{w}_T)]}{6\lambda T}\right) + \mathcal{O}(\eta^2\sigma_d^2). \tag{D.27}$$

This concludes the proof. $\qquad\square$

