# OpenReview forum: "Federated Learning with Projected Trajectory Regularization"
_TMLR — Accepted by TMLR_

### Review · Reviewer_zPZP · 2026-04-16

**Summary Of Contributions:**

> Summary:

This paper introduces FedPTR, a federated learning framework for for heterogeneous data that regularizes each client not toward the current global model (as it is done in FedProx), but toward a projected next-step model inferred from the recent global training trajectory. The main idea is to use Matching Training Trajectories to optimize an auxiliary synthetic dataset that mimics the recent global dynamics. Gradient steps on this auxiliary dataset then produce a projected point that serves as the proximal center. Convergence theorems and deep learning experiments are included to further support the validity of the method.

> Strengths:

The motivation of the paper is clear. I believe using a carefully designed estimate instead of the current iterate as the proximal center can lead to faster convergence. The high level idea is clear. The authors are trying to use past trajectory for this estimate, which makes sense. In fact, I recall there is a line of work in the acceleration of proximal / trust region method using extrapolated proximal center, which seems a bit similar in motivation as the current paper. The paper also provides a lot of empirical evidence, which further strengthen the claims made.

> Weaknesses:

1. The theory in the paper seems to rely on an assumption that the auxiliary loss gradient is close to the true global gradient. So the analysis is somewhat conditional on the success of the synthetic-data mechanism rather than explaining why it should succeed.
2. As I have mentioned the comparison with the acceleration using extrapolated center of (stochastic) proximal point method (aka FedProx) seems to be missing. I would suggest the authors at least mention this line of work, since the usage of an alternative proximal center itself is not a new idea.
3. The experiments are not comprehensive in the sense that all experiments are based on the comparison in terms of iteration complexity (number of communication rounds). This is not enough as the algorithm introduces substantial extra machinery through MTT and auxiliary-data optimization, but the paper does not provide a serious accounting of extra computation, memory, or wall-clock cost, nor does it compare methods under matched resource budgets. It is very likely that any improvement in iteration complexity comes at the cost of substantially increased local computation. In particular, I believe the authors should compare their method against a simple baseline that uses an extrapolated proximal center.

**Audience:**

Yes

**Audience Explanation:**

The idea of using the recent global trajectory rather than only the current global iterate is interesting, and the paper’s framing of synthetic data based trajectory projection as a regularization target is novel enough. Even if the current version does not fully close the case, it points to a direction that may be useful beyond this specific algorithm, which is that using richer temporal information from optimization trajectories to guide local updates.

**Claims And Evidence:**

Yes

**Claims Explanation:**

I would say partially yes. Some of claims require further justifications.

1. The convergence theorem depends on Assumption 5.3, which requires the auxiliary synthetic objective to have gradient close to the true global objective. This assumption is central and strong. As a result, the theory reads more like a conditional guarantee.

2. As I have mentioned previously, there are gaps in experiments.

**Requested Changes:**

I would suggest adding the following to the paper:

1. Discuss the line of work on accelerated proximal-point methods and proximal-center extrapolation in the background or related-work section.
2. Include experiments reporting the total amount of computation, not only the number of communication rounds, required to reach a given accuracy level.
3. While I not think it is absolutely necessary. Relaxing the assumptions used in the convergence theorem would make the theoretical results significantly more convincing.

---

> ### Author Response · Authors · 2026-05-18
> **Response to Reviewer zPZP (1)**
>
> Thank you for your valuable comments!
>
> **Q1:** Comparison with extrapolated proximal center, and related literature
>
> We thank the reviewer for pointing to the line of work on accelerated proximal-point methods and proximal-center extrapolation. To directly address this concern, we implemented a **FedProx + linear extrapolation** baseline, where the proximal center is set to $\text{anchor}^t = w^t + \rho \cdot (w^t - w^{t-1})$ (server-computed and pushed to all clients each round; falls back to $w^t$ in the first round). The client local objective then becomes $\arg\min_w L(w; D_i) + \frac{\lambda}{2} ||w - \text{anchor}^t||^2$. We tested two extrapolation strengths, $\rho=0.5$ (half-step) and $\rho=1.0$ (full-step), on CIFAR-10 with Dir($\alpha=0.01$). All other settings are identical to the main paper.
> | Method | Acc |
> |---|---|
> | FedProx | 55.94 |
> | FedProx + linear extrap ($\rho=0.5$) | 59.18 |
> | FedProx + linear extrap ($\rho=1.0$) | 57.73 |
> | FedPTR | 69.04 |
> Linear extrapolation provides a modest +3.24-point improvement at $\rho=0.5$ but degrades at $\rho=1.0$, since under extreme heterogeneity the aggregated trajectory direction $w^t - w^{t-1}$ is itself noisy and aggressive extrapolation amplifies that noise. FedPTR substantially outperforms the best linear-extrap baseline by **+9.86 points**, indicating that the gain comes not from anchor displacement per se but from a nonlinear, data-driven trajectory projection through MTT. We have also added a discussion of accelerated proximal-point methods and proximal-center extrapolation to the related-work section, and the corresponding empirical comparison to Appendix E of the revised paper.
>
> **Q2:** Memory and computation overhead.
>
> We thank the reviewer for raising this important concern. Below we provide a quantitative accounting of FedPTR's per-client memory and computation cost on CIFAR-10 with Dirichlet $\alpha=0.01$, measured on a single NVIDIA A6000.
>
> **Memory.** As summarized in the table below, beyond the standard global model $w$, FedPTR clients carry two additional persistent items: the projected next-step model $\tilde{w}_i^{t+1}$ ($1\times$ model size) and the auxiliary dataset $\tilde{\mathcal{D}}_i$. The size of $\tilde{\mathcal{D}}_i$ is a fixed hyperparameter independent of training horizon and model dimension. For CIFAR-10 with $|\tilde{\mathcal{D}}_i|=100$ images, it amounts to $1.17$ MB. For FedPTR-S, the auxiliary dataset is moved to the server, so the client-side overhead matches FedDyn's.
> | Method   | Per-client memory | Components |
> |--|--|--|
> | FedAvg   | 1.22 MB | $w$ only |
> | FedDyn   | 2.44 MB | $w$ + local gradient state |
> | FedPTR   | 3.61 MB | $w$ + projected next-step model + $\tilde{\mathcal{D}}_i$ (1.17 MB) |
> | FedPTR-S | 2.44 MB | $w$ + projected next-step model |
>
> **Computation.** As shown in the table below, per-client wall-clock per round is 1.26 s for FedAvg, 1.29 s for FedProx, and 6.26 s for FedPTR with MTT every round (a 4.97x overhead). However, MTT updates need not be performed every round. With reduced MTT frequency (paper Section 6.3), per-client wall-clock approaches FedAvg's while accuracy remains substantially above FedAvg/FedProx. In particular, FedPTR with one-shot MTT runs at essentially FedAvg's wall-clock (1.06x) while still outperforming all baselines in paper Table 2.
>
> | Method                       | Per-client wall-clock | x FedAvg | Acc |
> |------------------------------|-----------------------|----------|-----|
> | FedAvg                       | 1.26 s | 1.00x | 55.43 |
> | FedProx                      | 1.29 s | 1.02x | 55.94 |
> | FedPTR (MTT every round)     | 6.26 s | 4.97x | 69.04 |
> | FedPTR (MTT per 4 rounds)    | 2.51 s | 1.99x | 66.87 |
> | FedPTR (MTT per 10 rounds)   | 1.79 s | 1.42x | 66.20 |
> | FedPTR (MTT per 40 rounds)   | 1.43 s | 1.13x | 65.74 |
> | FedPTR (one-shot MTT)        | 1.33 s | 1.06x | 64.67 |
>
> We also report per-client wall-clock to first reach a given accuracy target below. Reduced-frequency FedPTR variants reach each target faster than FedAvg/FedProx (e.g. FedPTR one-shot reaches 50% in 63s vs FedAvg's 205s), and FedPTR is the only method able to reach 65%+ accuracy.
>
> | Method                    | time to 50% | time to 55% | time to 60% | time to 65% |
> |--|--|--|--|--|
> | FedAvg                    | 205 s | 353 s | N/A   | N/A   |
> | FedProx                   | 179 s | 311 s | N/A   | N/A   |
> | FedPTR (every round)      | 244 s | 344 s | 520 s | 983 s |
> | FedPTR (per 4 rounds)     | 103 s | 153 s | 233 s | 412 s |
> | FedPTR (per 10 rounds)    |  86 s | 125 s | 190 s | 394 s |
> | FedPTR (per 40 rounds)    |  67 s |  96 s | 173 s | 365 s |
> | FedPTR (one-shot)         |  63 s | 106 s | 176 s | 382 s |
>
> These memory and computation comparisons clearly demonstrate that FedPTR introduces only modest overhead while delivering substantially higher accuracy, fully validating its practical viability. We have added these experiments and analysis to Appendix D of the revised paper.

---

> ### Author Response · Authors · 2026-05-18
> **Response to Reviewer zPZP (2)**
>
> **Q3:** Regarding Assumption 5.3 and its relaxation.
> We thank the reviewer for this thoughtful comment. Upon reflection, we observe that the original deterministic form of Assumption 5.3 is stronger than strictly necessary for our convergence result, and we address this concern through both a relaxation of the assumption and a clarification of the intended scope of our analysis.
>
> * **Relaxation of Assumption 5.3.** In our revised manuscript, we relax Assumption 5.3 from its deterministic form to an expectation form, where the expectation is taken over the randomness in the auxiliary dataset $\tilde{D}_i$ produced by the MTT procedure:
> $E_D[|| \cdot ||_2^2] \leq \sigma_d^2$ (here $D$ should be $\tilde{D}_i$ due to format issue).
> Crucially, this relaxed assumption no longer presumes that *any individual* auxiliary objective uniformly captures the global training dynamics; it only requires the MTT procedure to approximate the global gradient *on average* over its stochastic outputs, permitting larger pointwise deviation for individual realizations of $\tilde{D}_i$. With this assumption relaxation, the convergence proof (Appendix H) requires only a minor modification: every invocation of Assumption 5.3 in the proof already appears within an expectation, so we simply take an additional expectation over the MTT randomness $\tilde{D}_i$ nested within the existing expectation over $t$. The relaxed assumption then suffices to recover Theorem 5.4 and Corollary 5.6 verbatim.
> * Regarding the concern that the analysis is conditional on the success of the synthetic-data mechanism rather than explaining why it should succeed.
> We respectfully note that explaining why MTT produces auxiliary datasets satisfying Assumption 5.3 is not the goal of our theoretical analysis. Our theory characterizes how FedPTR's convergence depends on the quality of the auxiliary gradient (quantified by $\sigma_d^2$). Analyzing the MTT optimization procedure itself remains an open problem in the dataset distillation literature and constitutes a significant standalone direction beyond the scope of the present work.
>
> Additional discussion related to Assumption 5.3 has been added to Appendix G of our revised manuscript.

---

### Review · Reviewer_LLHg · 2026-04-19

**Summary Of Contributions:**

**Summary:** The paper proposes FedPTR, a federated learning (FL) framework designed to address data heterogeneity across clients. The method departs from existing approaches that rely on the current global model or linear combinations of past updates, and instead leverages recent global training trajectories to extract information for guiding local optimization. Specifically, the method employs Matching Training Trajectories (MTT) to construct an auxiliary synthetic dataset that approximates the recent dynamics of global model updates. This auxiliary dataset is then used to perform several gradient steps starting from the current global model, producing a projected next-step model. Local clients incorporate this projected model into their objective via a proximal regularization term, encouraging updates to align with the inferred trajectory direction. In addition, the paper introduces a layer-adaptive regularization parameter to account for differences across model layers, and presents a server-side variant (FedPTR-S) that centralizes the auxiliary dataset optimization to reduce client-side computation. The authors provide a convergence analysis under standard assumptions for non-convex stochastic optimization and demonstrate an $O(1/T)$ convergence rate. Experimental results on multiple image classification benchmarks under non-i.i.d. settings show improvements over several existing FL methods.

**Strengths:**
- This paper proposes a trajectory-based mechanism for incorporating global information, which differs from existing point-wise or linear-history approaches.
- This paper introduces a novel use of dataset distillation techniques (MTT) within federated learning to approximate training dynamics.
- It includes theoretical analysis with convergence guarantees under commonly used assumptions and demonstrates empirical improvements across multiple datasets and heterogeneity settings.

**Weaknesses:**
- A primary concern is the computational complexity of the proposed method. FedPTR introduces extra components such as MTT, auxiliary dataset optimization, and projected updates, which are significantly more expensive than standard FL methods. While the paper includes ablations on reducing update frequency, it does not provide quantitative analysis of training time, computational overhead, or memory usage. Thus, it is unclear whether the observed performance gains justify the additional cost, especially in resource-constrained federated settings.
- The effectiveness of the approach depends on the quality of the auxiliary dataset in approximating global training dynamics, which is not explicitly validated beyond indirect measures.
- Experimental evaluation is primarily limited to image classification tasks, and the applicability to other domains is unclear.

**Audience:**

Yes

**Audience Explanation:**

At least some individuals in TMLR's audience would likely be interested in this work. The paper addresses a central data heterogeneity challenge in FL and proposes a novel trajectory-based regularization approach that differs from existing methods. Researchers working on federated optimization, distributed learning, and dataset distillation may find the idea of leveraging training trajectories through synthetic data particularly relevant.

**Broader Impact Concerns:**

The paper does not raise significant new ethical concerns beyond those already associated with standard FL.

**Claims And Evidence:**

No

**Claims Explanation:**

The paper provides empirical evidence showing consistent accuracy improvements over several strong FL baselines across multiple benchmark datasets and heterogeneous data settings. It also includes ablation studies to validate the impact of key components, presents a theoretical convergence analysis under standard assumptions, and offers some mechanism-level evidence (e.g., gradient similarity) to support the intuition behind the proposed method.

However, several claims are not fully supported by the provided evidence. In particular, the paper does not include a quantitative analysis of computational or system overhead, nor does it evaluate the cost-performance trade-off under fixed computational budgets. Additionally, the experiments are limited to vision tasks, leaving the generality of the approach to other domains unclear. Finally, the assumption that the auxiliary dataset accurately captures global training dynamics is only indirectly validated and not systematically examined across different settings.

**Requested Changes:**

- The paper introduces additional components such as MTT optimization, auxiliary dataset maintenance, and trajectory projection, which likely incur significant computational cost. The submission should include comparisons of runtime, FLOPs, memory usage, and/or client–server workload against strong baselines. Without this, it is difficult to assess the practical viability of the method.
- All current comparisons are based on final accuracy. It would strengthen the claims to include experiments under equal training time or computational constraints to demonstrate whether the proposed method provides a favorable cost–performance trade-off.
- The method relies on the assumption that the synthetic dataset accurately captures global training dynamics. While some indirect evidence is provided, more systematic validation (e.g., across datasets, training stages, or different heterogeneity levels) would strengthen the justification.
- Experiments are limited to image classification benchmarks. Including results on other modalities (or at least a more detailed discussion of applicability and limitations) would improve the generality of the work.

---

> ### Author Response · Authors · 2026-05-18
> **Response to Reviewer LLHg (1)**
>
> Thank you for your valuable comments!
>
> **Q1:** Memory and computation overhead.
>
> We thank the reviewer for raising this important concern. Below we provide a quantitative accounting of FedPTR's per-client memory and computation cost on CIFAR-10 with Dirichlet $\alpha=0.01$, measured on a single NVIDIA A6000.
>
> **Memory.** As summarized in the table below, beyond the standard global model $w$, FedPTR clients carry two additional persistent items: the projected next-step model $\tilde{w}_i^{t+1}$ ($1\times$ model size) and the auxiliary dataset $\tilde{\mathcal{D}}_i$. The size of $\tilde{\mathcal{D}}_i$ is a fixed hyperparameter independent of training horizon and model dimension. For CIFAR-10 with $|\tilde{\mathcal{D}}_i|=100$ images, it amounts to $1.17$ MB. For FedPTR-S, the auxiliary dataset is moved to the server, so the client-side overhead matches FedDyn's.
>
> | Method   | Per-client memory | Components |
> |----------|-------------------|------------|
> | FedAvg   | 1.22 MB | $w$ only |
> | FedDyn   | 2.44 MB | $w$ + local gradient state |
> | FedPTR   | 3.61 MB | $w$ + projected next-step model + $\tilde{\mathcal{D}}_i$ (1.17 MB) |
> | FedPTR-S | 2.44 MB | $w$ + projected next-step model |
>
> **Computation.** As shown in the table below, per-client wall-clock per round is 1.26 s for FedAvg, 1.29 s for FedProx, and 6.26 s for FedPTR with MTT every round (a 4.97x overhead). However, MTT updates need not be performed every round. With reduced MTT frequency (paper Section 6.3), per-client wall-clock approaches FedAvg's while accuracy remains substantially above FedAvg/FedProx. In particular, FedPTR with one-shot MTT runs at essentially FedAvg's wall-clock (1.06x) while still outperforming all baselines in paper Table 2.
>
> | Method                       | Per-client wall-clock | x FedAvg | Acc |
> |------------------------------|-----------------------|----------|-----|
> | FedAvg                       | 1.26 s | 1.00x | 55.43 |
> | FedProx                      | 1.29 s | 1.02x | 55.94 |
> | FedPTR (MTT every round)     | 6.26 s | 4.97x | 69.04 |
> | FedPTR (MTT per 4 rounds)    | 2.51 s | 1.99x | 66.87 |
> | FedPTR (MTT per 10 rounds)   | 1.79 s | 1.42x | 66.20 |
> | FedPTR (MTT per 40 rounds)   | 1.43 s | 1.13x | 65.74 |
> | FedPTR (one-shot MTT)        | 1.33 s | 1.06x | 64.67 |
>
> We also report per-client wall-clock to first reach a given accuracy target below. Reduced-frequency FedPTR variants reach each target faster than FedAvg/FedProx (e.g. FedPTR one-shot reaches 50% in 63s vs FedAvg's 205s), and FedPTR is the only method able to reach 65%+ accuracy.
>
> | Method                    | time to 50% | time to 55% | time to 60% | time to 65% |
> |---------------------------|-------------|-------------|-------------|-------------|
> | FedAvg                    | 205 s | 353 s | N/A   | N/A   |
> | FedProx                   | 179 s | 311 s | N/A   | N/A   |
> | FedPTR (every round)      | 244 s | 344 s | 520 s | 983 s |
> | FedPTR (per 4 rounds)     | 103 s | 153 s | 233 s | 412 s |
> | FedPTR (per 10 rounds)    |  86 s | 125 s | 190 s | 394 s |
> | FedPTR (per 40 rounds)    |  67 s |  96 s | 173 s | 365 s |
> | FedPTR (one-shot)         |  63 s | 106 s | 176 s | 382 s |
>
> These memory and computation comparisons clearly demonstrate that FedPTR introduces only modest overhead while delivering substantially higher accuracy, fully validating its practical viability. We have added these experiments and analysis to Appendix D of the revised paper.

---

> ### Author Response · Authors · 2026-05-18
> **Response to Reviewer LLHg (2)**
>
> **Q2:** Auxiliary dataset quality across settings.
>
> We thank the reviewer for this suggestion. In paper Section 6.5 we already provide quantitative evidence on the quality of the extracted global information. Specifically, we approximate the global gradient by the server update direction $w^t - w^{t+1}$, approximate the auxiliary gradient by $w^t - \tilde{w}^{t+1}_i$, and the local gradient by $w^t - w^{t+1}_i$, then directly measure the cosine similarity between (a) the auxiliary gradient and the global gradient, and (b) the local gradient and the global gradient. To avoid the auxiliary dataset influencing the training gradient itself, training proceeds with vanilla FedAvg (no proximal term), while clients still update the auxiliary dataset and the projected trajectory on the side. Paper Figure 3 reports this for one randomly chosen client on CIFAR-100 with Dir(0.01), showing that the auxiliary gradient is consistently far closer to the global gradient than the local gradient throughout training. To further validate this across settings, we extend the same gradient-similarity analysis to CIFAR-10 with Dir($\alpha=0.01$) and Dir($\alpha=0.04$). For each setting we report the highest-5 cosine similarity between (a) the auxiliary gradient and the global gradient, and (b) the local gradient and the global gradient:
>
> | Setting | $\cos(g_{\text{aux}}, g_{\text{global}})$ | $\cos(g_{\text{local}}, g_{\text{global}})$ | gap |
> |---|---|---|---|
> | CIFAR-10 Dir($\alpha=0.01$) | 0.392 | 0.065 | +0.328 |
> | CIFAR-10 Dir($\alpha=0.04$) | 0.552 | 0.234 | +0.318 |
>
> Across both heterogeneity levels, the auxiliary gradient is consistently far closer to the global gradient than the local gradient, confirming that FedPTR's MTT-extracted auxiliary dataset captures global training dynamics that the raw local gradient does not.
>
> **Q3:** Applicability beyond image classification.
>
> We thank the reviewer for raising this point. The current restriction of FedPTR to image tasks is not a limitation of the federated framework itself, but inherits from the underlying dataset-distillation backbone. Language inputs are sequences of discrete tokens, so the synthetic data cannot be directly optimized by gradient-based methods such as Matching Training Trajectories (MTT), which require differentiable input parameterization. Some recent work tries to sidestep token-level non-differentiability by distilling auxiliary signals such as soft labels [1] or attention labels [2], and these efforts remain limited to small models and short-text classification. In particular, no MTT-style trajectory matching method has been reported for text in recent work [3]. We view extending FedPTR to language and other modalities as an important future direction, which we expect to become tractable as trajectory-based dataset distillation matures beyond image inputs. We have added this discussion to Appendix F of the revised paper.
>
> [1] Sucholutsky, Ilia, and Matthias Schonlau. "Soft-label dataset distillation and text dataset distillation." 2021 International Joint Conference on Neural Networks (IJCNN). IEEE, 2021.
>
> [2] Maekawa, Aru, et al. "Dataset Distillation with Attention Labels for Fine-tuning BERT." Proceedings of the 61st Annual Meeting of the Association for Computational Linguistics (Volume 2: Short Papers). 2023.
>
> [3] Maekawa, Aru, et al. "DiLM: Distilling Dataset into Language Model for Text-level Dataset Distillation." Findings of the Association for Computational Linguistics: NAACL 2024.

---

### Review · Reviewer_LrsU · 2026-04-30

**Summary Of Contributions:**

The paper proposes FedPTR, a federated learning framework designed to handle data heterogeneity by leveraging information from global training trajectories to regularize local updates. The authors provide convergence analysis under stochastic non-convex settings and report an $O(1/T)$ rate. Experimental results on several benchmark datasets under non-i.i.d. scenarios are included to evaluate the method.

**Additional Comments:**

N/A

**Audience:**

Yes

**Audience Explanation:**

Yes. I believe some individuals in the TMLR audience, particularly those working on federated learning and optimization under data heterogeneity, would find the findings of this paper of interest.

**Broader Impact Concerns:**

I do not identify significant ethical concerns specific to this work.

**Claims And Evidence:**

Yes

**Claims Explanation:**

Strengths:

1. The paper is well-written and clearly presented.

2. The problem is well-motivated, particularly in the context of data heterogeneity in federated learning.

3. The proposed method is interesting and presents a novel perspective on leveraging training dynamics.

Recommendations:

1. I recommend expanding the discussion of more recent SOTA methods for handling non-i.i.d. data in federated learning, to better position the contribution of this work.

2. I am curious whether the parameter \beta is missing from the input of Algorithm 2, as it appears in the formulation but is not clearly specified in the algorithm.

3. I recommend including an ablation study on the choice of $m$ across different models and datasets, as this parameter seems important to the method but its impact is not sufficiently explored.

4. I am concerned that the method may require clients to maintain all past global model states, as well as to store and continuously update the auxiliary dataset locally. It would be helpful to clarify the memory and computation overhead in practice.

5. I am concerned that the theoretical analysis only provides convergence guarantees in terms of the expected gradient norm, without establishing trajectory-level guarantees. In particular, Assumption 5.3 requires the auxiliary gradient to be uniformly close to the global gradient across the parameter space, which appears quite strong and may not hold under realistic data heterogeneity or imperfect auxiliary datasets. This assumption seems to implicitly presume that the auxiliary objective already captures the global training dynamics. I am curious whether MMT can truly capture the (precise) model training trajectory as claimed when only the start and end models are provided.

6. I am concerned that the convergence bound includes a residual term $O(\eta^2 \sigma_d^2)$, but the analysis does not explicitly characterize how this error accumulates over multiple steps, especially when $t - m$ becomes large. It is unclear whether the method can faithfully track the global training trajectory over long horizons. I am curious about the capability of capturing trajectories as the time gap increases, which also seems partially reflected in Figure 3.

7. I am concerned that, in non-convex optimization, multiple trajectories may connect the same initialization and final model, and matching only the endpoints does not uniquely determine the trajectory. The current theory does not address this ambiguity or clarify which trajectory is being approximated. I am curious whether there could be many plausible trajectories in the optimization space.

8. I am curious why FedPTR (either local or global variant) does not achieve the Pareto frontier in the experiments. I recommend providing more insights or discussion to better explain these observations.

9. I am curious about the impact of different initialization strategies for the synthesized data. It would be helpful to include more discussion or experiments on this aspect.

**Requested Changes:**

Please see the Recommendations outlined above.

---

> ### Author Response · Authors · 2026-05-18
> **Response to Reviewer LrsU (1)**
>
> Thank you for your valuable comments!
>
> **Q1:** Recent SOTA discussion.
>
> We thank the reviewer for this suggestion. In the revision, we have expanded our discussion on tackling data heterogeneity in FL to include prototype-based and representation-based FL methods, as well as heterogeneous FL with large generative models.
>
> **Q2:**  $\beta$ missing from Algorithm 2 input.
>
> We thank the reviewer for catching this. $\beta$ is the inner-step learning rate
> used at line 5 of Algorithm 2. The revised Algorithm 2 input is updated to
> include $\beta$.
>
>
> **Q3:** Ablation study on $m$.
>
> We thank the reviewer for this suggestion. We extend the trajectory length ablation to four (dataset $\times$ heterogeneity) combinations covered by the main results, with $m \in \\{3, 5, 10 \\}$ and all other settings identical to the main paper:
>
> | Setting | $m=3$ (paper main) | $m=5$ | $m=10$ |
> |---|---|---|---|
> | CIFAR-10 Dir($\alpha=0.01$) | 69.04 | 69.83 | 70.21 |
> | CIFAR-10 Dir($\alpha=0.04$) | 72.78 | 70.85 | 71.09 |
> | CIFAR-100 Dir($\alpha=0.01$) | 44.98 | 44.27 | 44.40 |
> | CIFAR-100 Dir($\alpha=0.04$) | 45.87 | 46.20 | 46.33 |
>
> The variation across $m \in \\{3, 5, 10\\}$ is bounded within ~1.5 points across all four settings. Notably, in two of the four settings (CIFAR-10 Dir($\alpha=0.01$) and CIFAR-100 Dir($\alpha=0.04$)), $m=10$ outperforms the paper's $m=3$ default by 1.17 and 0.46 points respectively, while in the other two settings $m=10$ stays within 1.7 points of $m=3$. This shows that FedPTR is robust to the choice of $m$, and that MTT continues to extract meaningful trajectory information even at a longer lookback gap ($m=10$), with the longer trajectory sometimes providing additional gain. We have added these experiments to Appendix B.6 of the revised paper.

---

> ### Author Response · Authors · 2026-05-18
> **Response to Reviewer LrsU (2)**
>
> **Q4:** Memory and computation overhead.
>
> **(a) Memory overhead:** we thank the reviewer for the question and would like to clarify that
> our algorithm does NOT require clients to maintain all past global model
> states. The algorithm only needs the most recent (m+1) global model snapshots
> to be retained in a sliding buffer, and in practice this buffer can be
> maintained on the server side. Consequently, FedPTR clients only carry two
> additional persistent items beyond standard FL methods: the projected
> next-step model $\tilde w^{t+1}_i$ and the auxiliary dataset $\tilde{\mathcal D}_i$. We summarize exact byte-level measurements on CIFAR-10 below. Note that for
> FedPTR-S, even the auxiliary dataset is moved to the server, so client-side
> overhead matches FedDyn.
> | Method   | Per-client memory | Notes |
> |----------|-------------------|-------|
> | FedAvg   | 1.22 MB | $w$ only |
> | FedDyn   | 2.44 MB | $w$ + local gradient state |
> | FedPTR   | 3.61 MB | $w$ + projected next-step model + $\tilde{\mathcal D}_i$ (1.17 MB) |
> | FedPTR-S | 2.44 MB | $w$ + projected next-step model |
>
> **(b) Computation overhead.** We report per-client wall-clock per round on CIFAR-10 with Dir(0.01), measured on a single NVIDIA A6000. Per-client wall-clock per round is 1.26 s for FedAvg, 1.29 s for FedProx, and 6.26 s for FedPTR with MTT every round (a 4.97× overhead). However, MTT updates need not be performed every round. With reduced MTT frequency (paper Section 6.3), per-client wall-clock approaches FedAvg's while accuracy remains substantially above FedAvg/FedProx. In particular, FedPTR with one-shot MTT runs at essentially FedAvg's wall-clock (1.06×) while still beating all baselines in paper table2.
> | Method                       | Per-client wall-clock | × FedAvg | Acc  |
> |------------------------------|-----------------------|----------|--------------|
> | FedAvg                       | 1.26 s | 1.00× | 55.43 |
> | FedProx                      | 1.29 s | 1.02× | 55.94 |
> | FedPTR (MTT every round)     | 6.26 s | 4.97× | 69.04 |
> | FedPTR (MTT per 4 rounds)    | 2.51 s | 1.99× | 66.87 |
> | FedPTR (MTT per 10 rounds)   | 1.79 s | 1.42× | 66.20 |
> | FedPTR (MTT per 40 rounds)   | 1.43 s | 1.13× | 65.74 |
> | FedPTR (one-shot MTT)        | 1.33 s | 1.06× | 64.67 |
>
> We also report time-to-target accuracy below. Reduced-frequency FedPTR variants reach each target faster than FedAvg/FedProx (e.g. FedPTR one-shot reaches 50% in 63s vs FedAvg's 205s), and FedPTR is the only method able to reach 65%+ accuracy.
> | Method                    | time to 50% | time to 55% | time to 60% | time to 65% |
> |---------------------------|-------------|-------------|-------------|-------------|
> | FedAvg                    | 205 s | 353 s | N/A   | N/A   |
> | FedProx                   | 179 s | 311 s | N/A   | N/A   |
> | FedPTR (every round)      | 244 s | 344 s | 520 s | 983 s |
> | FedPTR (per 4 rounds)     | 103 s | 153 s | 233 s | 412 s |
> | FedPTR (per 10 rounds)    |  86 s | 125 s | 190 s | 394 s |
> | FedPTR (per 40 rounds)    |  67 s |  96 s | 173 s | 365 s |
> | FedPTR (one-shot)         |  63 s | 106 s | 176 s | 382 s |
>
> Overall, these memory and computation comparisons clearly demonstrate that FedPTR introduces only modest overhead while delivering substantially higher accuracy, fully validating its practical viability. We have added these experiments and analysis to Appendix D of the revised paper.

---

> ### Author Response · Authors · 2026-05-18
> **Response to Reviewer LrsU (3)**
>
> **Q5:** Regarding Assumption 5.3.
>
> We thank the reviewer for this insightful comment.
> * We agree that the original deterministic form of Assumption 5.3 is stronger than necessary. In our revised manuscript, we relax Assumption 5.3 to its expectation form, where the expectation is taken over the randomness in the auxiliary dataset $\tilde{D}_i$, which is produced by the MTT process:$E_D [|| \cdot||_2^2] \leq \sigma_d^2$ (here $D$ should be $\tilde{D}_i$ due to format issue). This relaxation is analogous to the standard bounded gradient variance treatment in prior FL literature. Crucially, the assumption no longer presumes that *any individual* auxiliary objective uniformly captures the global training dynamics, it only requires the MTT procedure to approximate the global gradient *on average* over its stochastic outputs, permitting larger pointwise deviation for individual realizations of $\tilde{D}_i$.
> * The convergence proof (Appendix H) requires only a slight modification: every invocation of Assumption 5.3 in the proof already appears within an expectation, so we simply take an additional expectation over the MTT randomness $\tilde{D}_i$ nested within the existing expectation. The relaxed assumption then suffices to recover Theorem 5.4 and Corollary 5.6 verbatim.
> * To empirically validate this relaxed assumption, we follow the experimental setup in Figure 3 (Section 6.5). Specifically, on the server side we use the global update direction $d_{global} = w^t - w^{t+1}$ as a proxy for the global gradient direction, and $d_{aux} = w^t - \frac{1}{N} \sum_i \tilde{w}^{t+1}$ (there is an $i$ in the subscript) as a proxy for the auxiliary gradient direction. We then measure  $||d_{global} - d_{aux}||$ throughout training as an empirical proxy for $\sigma_d$. We observe that the norm ranges from approximately 2-3 in early training rounds to 6-8 in the last few training rounds. This confirms that $\sigma_d^2$ remains bounded in practice, consistent with the Assumption 5.3.
> * Regarding the trajectory-level guarantees and the precision of MTT, we note that establishing strict trajectory-level convergence,  for a global optimum and optimal reference trajectory is generally intractable in non-convex stochastic optimization, since the optimization landscape admits multiple stationary points and trajectories are inherently non-unique. But similar to our proof, we can show that $E[||w^t- \tilde{w}^{t+1}||^2] = \eta^2 E[||\nabla \tilde{L}(w^t; \tilde{\mathcal{D}}_i)||^2]$, and $E[||\nabla \tilde{L}(w^t; \tilde{\mathcal{D}}_i)||^2]$ can be bounded by the gradient dissimilarity $\sigma_d^2$ and the global gradient norm $E[||\nabla L(w^t)||^2]$ (similar to Eq. (H.12) and (H.13)). We would also like to clarify our claim regarding MTT: FedPTR does not require MTT to recover the precise training trajectory. Under our relaxed Assumption 5.3, we only require the auxiliary gradient to approximate the global gradient in expectation. This is a substantially weaker requirement than full trajectory reconstruction, and is consistent with MTT's original design goal of matching endpoint behavior rather than intermediate states.
>
> **Q6:** Regarding the residual term and the trajectory.
>
> We thank the reviewer for raising this concern. We address the two aspects of this question separately.
> * On the relationship between $\sigma_d^2$ and the MTT time gap.We would like to clarify a subtle point about Assumption 5.3. The quantity $\sigma_d^2$ bounds the gradient dissimilarity between the auxiliary loss function $\tilde{L}(w; \tilde{D}_i)$ and the global objective function $L(w)$, evaluated at the same point $w$. Crucially, this is a property of the auxiliary dataset $\tilde{D}_i$ itself, not of the MTT procedure that generates it. The number of MTT inner steps, the trajectory time gap $m$ and any other algorithmic details of MTT all influence the quality of $\tilde{D}_i$ (and thus the value of $\sigma_d^2$ for a given run), but they do not enter Assumption 5.3 as separate accumulating quantities. In other words, $\sigma_d^2$ captures whether the final $\tilde{D}_i$ yields a useful auxiliary loss rather than how it was obtained step-by-step. Consequently, the residual $O(\eta^2 \sigma_d^2)$ in our convergence bound does not require characterization of multi-step error accumulation through MTT.
> * On long-horizon trajectory tracking. As part of our response to Q5, we conducted an empirical study measuring the difference between global update direction and the auxiliary update direction throughout the entire training horizon (added as a new figure in Appendix G). The fact that this quantity remains bounded across hundreds of rounds is itself direct empirical evidence that FedPTR faithfully tracks the global trajectory over long horizons.

---

> ### Author Response · Authors · 2026-05-18
> **Response to Reviewer LrsU (4)**
>
> **Q7:** Regarding the optimization trajectories.
>
> We thank the reviewer for raising this conceptual question. We address the concern as follows.
> * As we noted in our response to Q5, our framework does not claim to identify the canonical training trajectory, as the reviewer correctly observes, no such canonical trajectory exists in general non-convex landscapes. Rather, FedPTR uses the auxiliary dataset $\tilde{D}_i$ for producing a better update reference than the last-step global model $w_t$ (used by FedProx). The relevant question is therefore whether the selected auxiliary dataset yields a useful regularization signal for guiding the global model update. Formally, our framework only requires that MTT produces an auxiliary dataset whose induced gradient field is sufficiently aligned with the global gradient *in expectation*. We do not need to identify a unique trajectory; we only need MTT to find *one* auxiliary dataset that is useful for regularization. Our empirical results confirm this weaker requirement is met in practice.
> * Regarding the reviewer's specific concern that "the current theory does not address this ambiguity or clarify which trajectory is being approximated", we clarify that our theoretical framework operates at the *point-wise* gradient level rather than the trajectory level. Assumption 5.3 and Theorem 5.4 are formulated entirely in terms of the auxiliary gradient but not in terms of trajectory-level properties such as path shape or uniqueness. Trajectory ambiguity therefore does not enter our analysis as an unresolved issue.
> * We also want to mention that this analytical choice is aligned with standard non-convex convergence analysis: results for SGD, FedAvg, FedAdam and other FL methods aim to certify convergence to a stationary point (typically via bounds on $\min_t \mathbb{E}||\nabla \mathcal{L}(w_t)||^2$), without characterizing *which* trajectory the optimizer takes. Our analysis follows this established convention and characterizes the key property that FedPTR's convergence depends on: when the auxiliary gradient provides a useful local correction signal. Additional discussion related to Assumption 5.3 has been added to Appendix G of our revised manuscript.
>
> **Q8:** Pareto frontier exceptions.
>
> We thank the reviewer for this observation. Across the 22 (dataset × heterogeneity × participation) combinations in paper Table 2 and Table 3, FedPTR or FedPTR-S is the **best** method in 21/22 combinations, and at least one of the two appears in the **top-2** in **22/22** combinations. The only combination where neither variant is best is 40 clients / 25% participation / CIFAR-100 / Dir(0.04), where FedDC reaches 42.74 vs. FedPTR 41.90 (a 0.84-point gap). We attribute this single exception to the joint effect of two factors specific to that setting: (1) very low effective per-round participation (10 of 40 clients sampled each round) introduces sampling noise into the cross-round trajectory that FedPTR's MTT relies on (2) Dir($\alpha=0.04$) is the milder heterogeneity setting, where local drift is less severe and FedPTR's advantage is most pronounced under extreme heterogeneity. Consistent with this, FedPTR is the best method in **every** Dir($\alpha=0.01$) setting across both tables.
>
> **Q9:** Initialization strategies.
>
> In paper Appendix B.1 (Table 8) we already compared two initialization strategies (Gaussian init vs. our class-aware initialization using local real data when available) and observed a 1.54-point gain on CIFAR-10. Here we consider three additional strategies for a more complete picture: **zero init** (all-zero pixels), **uniform init** ($\mathcal{U}(-1,1)$ per pixel), and **local-random init** (each synthetic slot filled with a randomly sampled local real image, ignoring class labels, which isolates the contribution of class-image alignment). Results on CIFAR-10 / Dir($\alpha=0.01$):
>
> | Strategy | Acc |
> |---|---|
> | FedAvg  | 55.43 |
> | FedDyn  | 62.29 |
> | FedPTR  | 65.12 |
> | FedPTR - Gaussian init | 67.50 |
> | FedPTR - uniform init | 68.51 |
> | FedPTR - local-random init | 69.00 |
> | FedPTR - class-aware init | 69.04 |
>
> Zero initialization clearly underperforms (65.12), confirming that MTT requires a non-trivial starting point. The two pure-noise inits (Gaussian 67.50, uniform 68.51) confirm that FedPTR is insensitive to the specific noise distribution. Local-random (real local images without class matching) reaches 69.00, within 0.04 of class-aware (69.04). This indicates that the gain over noise-based initialization comes mainly from using real local images, not from matching classes, which is also consistent with FedPTR-S (server-side, Gaussian init only) still substantially outperforming all baselines. Across all five FedPTR variants, every initialization strategy, including zero init, outperforms FedDyn, showing that FedPTR's effectiveness is not sensitive to the choice of initialization. We have added these experiments and analysis to Appendix B.1 of the revised paper.

---

### Decision · Action_Editor_EHx5 · 2026-06-06

**Recommendation:** Accept as is

**Additional Comments:**

Please incorporate the reviewer's suggestions into the camera-ready version. In addition, the convergence rate is established for the full-gradient setting rather than the stochastic-gradient setting. It would be helpful to emphasize this point, as readers may otherwise be confused by the convergence result presented in Corollary 5.6.

**Audience:**

Yes

**Audience Explanation:**

The idea of using the recent global trajectory rather than only the current global iterate is interesting, and the paper’s framing of synthetic data based trajectory projection as a regularization target is novel, which has the potential to advance federated learning in heterogeneous data settings.

**Claims And Evidence:**

Yes

**Claims Explanation:**

This paper develops a new federated learning method to handle the non-identically distributed data across the clients. Particularly, it allows local clients or the server to optimize an auxiliary (synthetic) dataset that mimics the learning dynamics of the recent model update and utilizes it to project the next-step model trajectory for local training regularization. Both convergence rates and extensive experiments are included to support the validity of the method.